# Correspondence Cognitive Learning for Multi-Modal Object Re-Identification

Chao Su [1]   Shuying Li [1]   Ruitao Pu [1]   Dezhong Peng [1]   Zhenwen Ren [2]   Yuan Sun [3]

## Abstract

Multi-modal object Re-Identification (ReID) aims to retrieve the same object across different modalities by exploiting their complementary visual information. Recent advances leverage Multimodal Large Language Models (MLLMs) to generate descriptive textual annotations as auxiliary supervision. However, existing approaches usually adopt these generated texts directly, overlooking the varying correspondence degrees between visual and textual modalities. Such neglect may lead the model to treat strong- and weak-correspondence image–text pairs equally, limiting its ability to learn discriminative associations and hindering effective optimization. To overcome this limitation, we propose a Correspondence Cognitive Learning (CCL) framework that explicitly models the correspondence degree and facilitates a progressive learning process from easy to hard pairs. CCL is composed of two synergistic modules. The Correspondence-Guided Semantic Refinement (CGSR) module dynamically refines visual representations using text semantics according to the correspondence difficulty estimated from the previous epoch, thereby enhancing feature alignment under imperfect associations. The Cognitive-Driven Dynamic Optimization (CDDO) module presents a self-paced weighting mechanism that adaptively adjusts the optimization focus by emphasizing easy pairs at the early stage and gradually integrating harder ones as training evolves. Together, these modules enhance feature-level alignment and optimization adaptivity, yielding robust and discriminative multi-modal representations. Extensive experiments on three multi-modal object ReID bench-

marks demonstrate the superior performance of our method.

## 1. Introduction

Object re-identification (ReID) (Li et al., 2025; Wu et al., 2026; Yang et al., 2026) aims to retrieve specific targets across different camera views and serves as a core task in intelligent video analysis. In recent years, multi-modal object ReID (Li et al., 2026b;c; Wan et al., 2026a;b) has achieved remarkable progress by integrating complementary information from heterogeneous modalities such as RGB, near-infrared (NIR), and thermal infrared (TIR). This integration effectively overcomes the limitations of single-modality imaging under challenging conditions such as darkness or strong illumination. For instance, DeMo (Wang et al., 2025b) skillfully addresses the issues of dynamic quality variation and information interference among multi-modal images through feature decoupling and mixture-of-experts mechanisms, pushing the performance of multi-modal ReID to new heights. With the rapid advancement of multi-modal large language models (MLLMs), researchers have begun to explore their potential in generating descriptive textual annotations for images (Wang et al., 2025c), aiming to construct image–text pairs that provide auxiliary semantic supervision for the ReID task. This text-augmented paradigm enriches the semantic context and offers valuable complementary information beyond visual cues. Recent advances (Su et al., 2025; 2026b;a) have shown that robust cross-modal consistency learning is crucial under weak supervision settings. However, existing multi-modal object ReID approaches usually adopt the generated texts directly, overlooking the varying correspondence degrees between visual and textual modalities. As shown in Figure 1, such neglect leads the model to treat strong- and weak-correspondence image–text pairs equally, limiting its ability to learn discriminative associations and hindering effective optimization.

This limitation gives rise to two correspondence-related challenges: (1) Existing methods fail to adaptively refine visual features according to correspondence degree. Without correspondence-aware guidance, textual semantics cannot effectively enhance visual representations, especially for weak-correspondence pairs. (2) Current optimization treats all pairs equally, ignoring their learning difficulty and reli-

[1]College of Computer Science, Sichuan University, Chengdu, China [2]Southwest University of Science and Technology, Mianyang, China [3]National Key Laboratory of Fundamental Algorithms and Models for Engineering Numerical Simulation, Sichuan University, Chengdu, China. Correspondence to: Yuan Sun <sun-yuan_work@163.com>.

*Proceedings of the 43rd International Conference on Machine Learning*, Seoul, South Korea. PMLR 306, 2026. Copyright 2026 by the author(s).

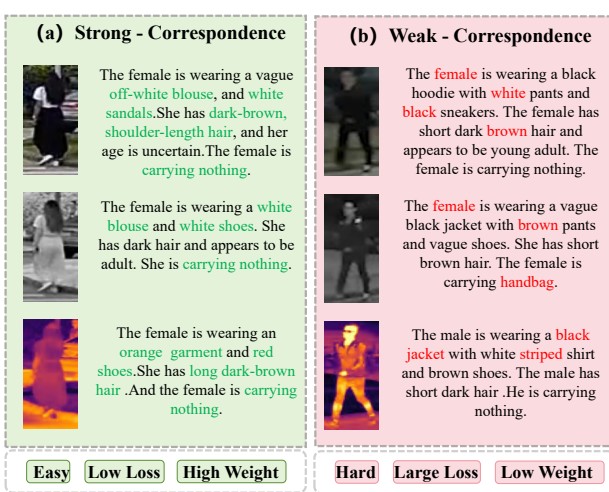

**Figure 1.** Cross-sample comparison of image–text correspondence degrees. (a) Strong-correspondence pair: clothing categories, colors, and accessories of the person (top, pants, shoes, carried items) are accurately aligned with the textual description across all spectral modalities. (b) Weak-correspondence pair: these key attributes visibly deviate from the text.

ability. This uniform strategy hinders progressive learning from strong- to weak-correspondence pairs and results in unstable convergence.

To tackle these challenges, we propose the Correspondence Cognitive Learning (CCL) framework, which explicitly models the correspondence degree and enables progressive, correspondence-aware learning for multi-modal ReID. CCL consists of two synergistic modules: Correspondence-Guided Semantic Refinement (CGSR) and Cognitive-Driven Dynamic Optimization (CDDO).

The CGSR module addresses the feature-level challenge by dynamically refining visual representations with textual semantics according to correspondence difficulty estimated from the previous epoch. By adaptively incorporating fine-grained text information, CGSR enhances representation quality for strong-correspondence pairs while mitigating the impact of weak-correspondence ones. The CDDO module tackles the optimization-level challenge via a cognitive-inspired self-paced strategy. It assigns adaptive weights to pairs based on their loss values, enabling the model to focus on easy pairs during early training and progressively include harder pairs as learning evolves. Through this dual design, CCL enhances both feature-level correspondence modeling and optimization-level adaptivity, leading to more robust and discriminative multi-modal representations. The main contributions of this work are summarized as follows:

- We propose a novel Correspondence Cognitive Learning (CCL) framework that explicitly models the correspondence degree between image–text pairs and

achieves progressive correspondence-aware learning for multi-modal object ReID.

- We design a Correspondence-Guided Semantic Refinement (CGSR) module that dynamically refines visual features with text semantics according to correspondence difficulty, addressing the feature-level correspondence challenge.

- The proposed Cognitive-Driven Dynamic Optimization (CDDO) module applies a self-paced weighting mechanism to adaptively adjust optimization focus, addressing the optimization-level correspondence challenge.

- Extensive experiments on three multi-modal object ReID benchmarks demonstrate that CCL consistently surpasses state-of-the-art methods in both performance and robustness.

## 2. Related Work

### 2.1. Multi-modal Object Re-Identification

Object re-identification (ReID) (Yan et al., 2023; 2024; Pang et al., 2025; Yu et al., 2025a; Deng et al., 2025; Gong et al., 2026) aims to retrieve the same target across different camera views and has long been a fundamental task in intelligent video analysis. Early studies mainly focus on learning discriminative visual representations to handle variations in viewpoint, illumination, and occlusion. To further enhance robustness in complex environments, multi-modal object ReID has emerged, which integrates complementary cues from heterogeneous modalities such as RGB, near-infrared (NIR), and thermal infrared (TIR), thereby mitigating the limitations of single-modality imaging. Recent advances (Wang et al., 2025b; Feng et al., 2025; Li et al., 2026a) have further improved multi-modal ReID by introducing mixture-of-experts (MoE) (Jacobs et al., 1991) mechanisms to achieve adaptive and instance-aware fusion across modalities. More recently, with the rapid progress of multi-modal large language models (MLLMs), researchers have begun incorporating textual information into ReID tasks to provide richer semantic supervision (Wang et al., 2025c).

However, existing text-augmented approaches often ignore the varying image–text correspondence, leading to suboptimal feature refinement when textual guidance is weak.

### 2.2. Self-paced Learning

Self-paced learning (SPL) (Kumar et al., 2010; Jiang et al., 2015) was originally proposed to mimic the human learning process of progressing from easy to hard samples, where sample difficulty is typically quantified by loss magnitude. This paradigm has been widely adopted to enhance model

robustness under noisy or uncertain supervision, as it allows the network to first learn from reliable samples and gradually incorporate more challenging ones. Most prior SPL work (Meng et al., 2017) assumes a single (intra-)modal data distribution and homogeneous semantic consistency among samples; such an assumption can be violated in multi-modal settings where supervision quality depends strongly on cross-modal correspondence. Recent efforts have extended SPL ideas to multi-modal and weakly-supervised scenarios (Sun et al., 2024; Pu et al., 2025), demonstrating that sample-selection rules must account for view-consistency or correspondence degree to avoid unstable training and degraded generalization.

This gives rise to our second challenge, which is how to achieve cognitive-driven dynamic optimization that progressively balances learning across samples with different correspondence degrees.

## 3. Methods

### 3.1. Correspondence-Guided Semantic Refinement

Existing methods (Wang et al., 2025c) incorporate auxiliary text descriptions without assessing their semantic reliability. However, the degree of image-text correspondence is non-uniform across samples. Strong-correspondence pairs exhibit high consistency, whereas weak-correspondence pairs often introduce ambiguity. To address this, we propose the Correspondence-Guided Semantic Refinement (CGSR) module. This module incorporates a cognitive-inspired mechanism that dynamically modulates the integration of textual semantics based on instance-level correspondence difficulty. By mimicking the human cognitive process of selective attention, this mechanism ensures that reliable semantic cues are prioritized while misleading signals from mismatched descriptions are suppressed.

**Cognitive Difficulty Estimation.** To effectively quantify the reliability of cross-modal correspondence, we introduce a difficulty estimation mechanism. Following the training pipeline of DeMo (Wang et al., 2025b), we initiate a warm-up phase for the first $E_{\text{warm}}$ epochs to accumulate optimization feedback. Formally, let $\mathcal{X} = \{x_i\}_{i=1}^K$ denote the training set with $K$ samples. For each sample $x_i$, we compute a composite feedback signal $\mathcal{L}_i$ at each epoch:

$$\mathcal{L}_i = \mathcal{L}_{\text{ce}}(x_i) + \mathcal{L}_{\text{tri}}(x_i), \tag{1}$$

where $\mathcal{L}_{\text{ce}}$ and $\mathcal{L}_{\text{tri}}$ represent the cross-entropy loss (Szegedy et al., 2016) and triplet loss (Hermans et al., 2017), respectively. We note that the estimated difficulty reflects optimization inconsistency caused by unreliable cross-modal supervision. Strong image-text correspondence typically provides stable auxiliary guidance and thus yields faster convergence and lower loss, whereas

weak-correspondence pairs often introduce semantic conflicts and lead to unstable trajectories with persistently larger losses during early training.

Starting from the first epoch after warm-up, we update a global loss vector $\mathbf{\Omega} \in \mathbb{R}^K$ at each epoch using the losses from the previous epoch. Subsequently, a normalized difficulty indicator $\mathbf{d} \in [0, 1]^K$ is derived via Min-Max normalization. Each element $d_i$ of $\mathbf{d}$ denotes the sample's relative difficulty. A higher $d_i$ indicates a harder sample, suggesting that the associated textual guidance may be less informative.

**Text-Guided Feature Refinement.** To fully exploit discriminative information from heterogeneous spectral modalities, parameter-shared CLIP Vision Transformers are employed for feature extraction. Specifically, we consider three modalities $m \in \{\text{R}, \text{N}, \text{T}\}$, denoting RGB, NIR, and TIR, respectively. For each modality, the visual encoder processes input images to generate a sequence of visual tokens $\mathbf{F}_v^m \in \mathbb{R}^{B \times N_p \times D}$ and a global class token $\mathbf{f}_{cls,v}^m \in \mathbb{R}^{B \times D}$. Here, $B$, $N_p$, and $D$ denote the batch size, the number of patch tokens, and the feature dimension, respectively. In parallel, the frozen CLIP text encoder processes modality-specific descriptions, yielding textual token embeddings $\mathbf{F}_t^m \in \mathbb{R}^{B \times L \times D}$ and a global semantic representation $\mathbf{f}_{cls,t}^m \in \mathbb{R}^{B \times D}$, where $L$ indicates the sequence length of textual tokens.

Building upon these extracted representations, we aim to establish fine-grained semantic alignment through a refinement mechanism inspired by previous work (Yang et al., 2022). Specifically, we estimate per-token scores to quantify the correspondence between the visual tokens and textual embeddings. The refinement process begins by injecting linguistic context via Cross-Attention (CA), with visual tokens as queries and textual embeddings as keys and values:

$$\mathbf{F}_{mix}^m = \mathbf{F}_v^m + \text{CA}(\mathbf{F}_v^m, \mathbf{F}_t^m, \mathbf{F}_t^m). \tag{2}$$

This operation enriches the visual representation with complementary textual cues. To explicitly verify the reliability of the injected semantics, we generate a semantic consistency mask. For the $j$-th token $\mathbf{f}_{mix,j}^m$ in $\mathbf{F}_{mix}^m$, we reuse the same attention mechanism to perform a secondary query over the text features, yielding a corresponding semantic anchor $\mathbf{s}_j^m$. This anchor serves as a reference prototype to quantify the semantic consistency of each visual token. To accurately measure the alignment, we project both vectors into a normalized common space:

$$\tilde{\mathbf{f}}_j^m = \frac{W_v \mathbf{f}_{mix,j}^m}{\|W_v \mathbf{f}_{mix,j}^m\|_2}, \quad \tilde{\mathbf{s}}_j^m = \frac{W_t \mathbf{s}_j^m}{\|W_t \mathbf{s}_j^m\|_2}, \tag{3}$$

where $W_v$ and $W_t$ are learnable projection matrices. The confidence scores are then formulated using a Gaussian

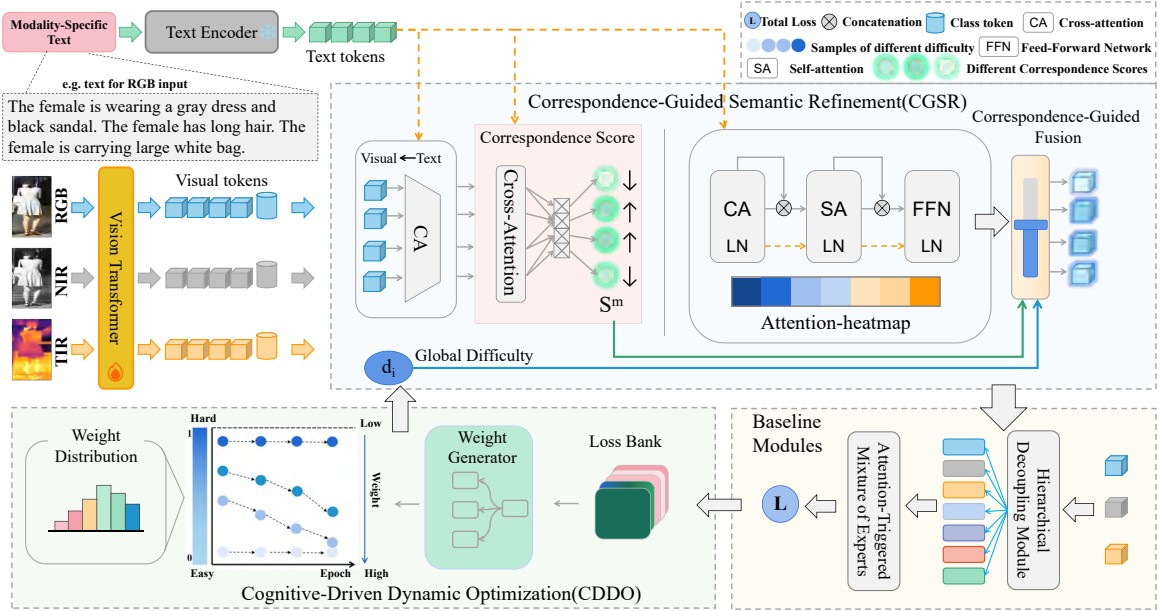

*Figure 2.* The pipeline of the proposed CCL. First, the Correspondence-Guided Semantic Refinement (CGSR) module refines visual features by injecting textual semantics under the guidance of instance-level correspondence difficulty. Next, the CDDO module assigns adaptive weights to image–text pairs according to their loss values, allowing the model to emphasize easy pairs at the beginning and gradually incorporate harder ones as training proceeds. Additionally, the Hierarchical Decoupling and Attention-Triggered MoE modules from DeMo baseline are kept for modality-specific learning and expert fusion.

kernel based on the cosine distance:

$$S_j^m = \alpha \cdot \exp\left(-\frac{(1 - \langle \tilde{\mathbf{f}}_j^m, \tilde{\mathbf{s}}_j^m \rangle)^2}{2\sigma^2}\right), \qquad (4)$$

where $\alpha$ and $\sigma$ are learnable scaling and bandwidth parameters, respectively. High $S_j^m$ values highlight tokens that are semantically consistent with the text. Conversely, low values identify misaligned or noisy regions. These scores collectively form the semantic mask $\mathbf{S}^m \in \mathbb{R}^{B \times N_p \times 1}$, which serves as a reliability gate for the subsequent step.

**Contextual Refinement.** Subsequently, we introduce a text-guided contextual refinement process that adaptively fuses cross-modal cues while preserving structural coherence among tokens. The enriched visual tokens interact again with textual semantics and with each other to construct a context-enriched representation:

$$\hat{\mathbf{F}}_v^m = \text{SA}\big(\mathbf{F}_{mix}^m + \text{CA}(\mathbf{F}_{mix}^m, \mathbf{F}_t^m)\big), \qquad (5)$$

where $\text{SA}(\cdot)$ denotes self-attention equipped with relative positional encoding to capture global structural dependencies across patches. The resulting features then undergo layer normalization and a feed-forward transformation with residual connections:

$$\tilde{\mathbf{F}}_v^m = \text{FFN}\big(\text{LN}(\hat{\mathbf{F}}_v^m)\big) + \hat{\mathbf{F}}_v^m, \qquad (6)$$

where LN and FFN denote Layer Normalization and Feed-Forward Network, respectively. Finally, the derived corre-

spondence scores $\mathbf{S}^m$ are employed to softly modulate these refined features to emphasize reliable semantics:

$$\mathbf{F}_{ref}^m = \tilde{\mathbf{F}}_v^m \odot \mathbf{S}^m, \qquad (7)$$

where $\odot$ denotes element-wise multiplication along the token dimension. Through this operation, $\mathbf{F}_{ref}^m$ effectively encodes verified textual semantics while suppressing visual tokens with weak correspondence.

**Correspondence-Guided Fusion.** Finally, to mitigate the risk of negative transfer originating from mismatched text, we employ a difficulty-aware gating mechanism. Specifically, the original visual features $\mathbf{F}_v^m$ and the text-refined features $\mathbf{F}_{ref}^m$ are adaptively fused at the sample level based on the sample difficulty:

$$\mathbf{F}^m = d_i \cdot \mathbf{F}_v^m + (1 - d_i) \cdot \mathbf{F}_{ref}^m. \qquad (8)$$

Under this formulation, the model explicitly prioritizes the refined features $\mathbf{F}_{ref}^m$ for easy samples (low $d_i$) while conservatively reverting to the original visual representation $\mathbf{F}_v^m$ for hard samples to ensure robustness.

Crucially, our framework establishes a hierarchical synergy between local verification and global gating. While the token-level mask $\mathbf{S}^m$ explicitly handles spatial inconsistencies by suppressing irrelevant local regions, relying exclusively on local alignment is risky for hard samples where text descriptions might be globally unreliable. Therefore,

the global indicator $d_i$ acts as a holistic reliability gate. It determines the overall trust placed in the text-refined features, ensuring that spatially aligned cues are incorporated strictly in proportion to the robustness of the sample-level correspondence. The fused representations $\mathbf{F}^m$, now endowed with better semantic coherence and cleaner signals, are provided as refined inputs to the Hierarchical Decoupling Module of DeMo (Wang et al., 2025b), thereby facilitating the learning of more pure and disentangled feature factors.

## 3.2. Cognitive-Driven Dynamic Optimization

While CGSR enhances feature representations through difficulty-aware semantic guidance, the optimization process still suffers from weak-correspondence samples. Inspired by human cognitive learning patterns where easy concepts are learned first, we introduce the Cognitive-Driven Dynamic Optimization (CDDO) module. This design is grounded in the "memorization effect" of Deep Neural Networks (DNNs) (Arpit et al., 2017), where models tend to prioritize learning simple, correctly aligned patterns before fitting complex or noisy samples. Empirical studies (Qin et al., 2022; 2024; He et al., 2026) demonstrate that this phenomenon results in a distinct loss distribution: clean pairs with strong correspondence yield relatively low loss values, whereas mismatched or weak-correspondence pairs exhibit significantly higher losses during the early stages of training. By leveraging this, CDDO dynamically re-weights the loss based on each sample's learning difficulty to ensure more robust visual-semantic alignment.

After the warm-up phase, given the sample losses $\{l_i\}_{i=1}^N$ computed from the total objective $\mathcal{L}$, we introduce a *self-paced weight* $w_i$ for each sample:

$$w_i = \exp\left(-\frac{l_i}{\rho}\right), \tag{9}$$

where $\rho$ is a temperature-like hyperparameter controlling the sensitivity to the loss magnitude, it is set to the $\gamma$-quantile from the global loss pool of the final warm-up epoch and kept fixed thereafter. Strong-correspondence (easy) samples with small $l_i$ obtain larger $w_i$, while weak-correspondence (hard) samples are softly suppressed. The dynamically weighted batch loss is then formulated as:

$$\mathcal{L}_{cddo} = \frac{1}{N}\sum_{i=1}^N w_i \cdot l_i + \frac{1}{N}\sum_{i=1}^N \rho(w_i - w_i ln(w_i)), \tag{10}$$

where $\frac{1}{N}\sum_{i=1}^N \rho(w_i - w_i ln(w_i))$ is the self-paced regularizer which acts as a soft gatekeeper that penalizes the inclusion of too many samples at once, thereby enforcing a curriculum-driven learning trajectory from easy to hard instances. More theoretical justification and analysis of the effectiveness of CDDO are provided in Appendix Sec. A.

## 3.3. Objective Functions

To achieve stable and progressive optimization, the overall loss $\mathcal{L}_{\text{total}}$ is defined as a two-phase piecewise function:

$$\mathcal{L}_{\text{total}} = \begin{cases} \mathcal{L}_{\text{cross}} + \mathcal{L}_{\text{triplet}}, & \text{if } e \leq E_{\text{warm}}, \\ \mathcal{L}_{cddo}, & \text{if } e > E_{\text{warm}}. \end{cases} \tag{11}$$

Here, the warm-up stage ($e \leq E_{\text{warm}}$) uses standard supervision to stabilize feature learning and estimate sample difficulty. Afterward, the cognitive-driven dynamic optimization phase ($e > E_{\text{warm}}$) applies dynamically weighted losses in $\mathcal{L}_{cddo}$, enabling the model to focus on strong-correspondence (easy) samples first and progressively incorporate weak-correspondence (hard) ones.

## 4. Experiments

### 4.1. Datasets and Evaluation Protocols

**Datasets.** In this work, we evaluate the effectiveness of CCL on the text-augmented versions of the RGBNT201 (Zheng et al., 2021), RGBNT100 (Li et al., 2020), and MSVR310 (Zheng et al., 2023b) datasets, following the protocol established in the prior work (Wang et al., 2025c). All textual descriptions are directly adopted from the annotations generated by IDEA (Wang et al., 2025c) using the Qwen-VL (Bai et al., 2023) model. Specifically, RGBNT201 is a multi-modal person ReID dataset containing 4,787 aligned RGB, NIR, and TIR triplets from 201 identities. For the text-augmented setting, we use 14,361 textual annotations, each averaging 35.4 characters and describing up to 8 discriminative attributes. RGBNT100 is a large-scale vehicle dataset with 17,250 triplets and 51,750 corresponding descriptions, each averaging 56.3 characters and annotated with 6 key attributes. Finally, MSVR310 focuses on multi-modal vehicle ReID and includes 2,087 image triplets with 6,261 auto-generated captions, averaging 56.1 characters and covering 6 semantic attributes.

**Evaluation Protocols.** The performance is evaluated via mean Average Precision (mAP) and Cumulative Matching Characteristics (CMC) at Rank-K (K = 1, 5, 10).

### 4.2. Implementation Details

We employ the CLIP-pretrained (Radford et al., 2021) visual and textual encoders as the backbone networks. Notably, textual data are only utilized during training, and no textual descriptions are required during inference, in accordance with the standard visual ReID paradigm. The input image resolution is set to $256 \times 128$ for RGBNT201, while it is set to $128 \times 256$ for RGBNT100 and MSVR310. For data augmentation, we adopt commonly used strategies in the ReID domain, including random horizontal flipping, random cropping, and random erasing (Zhong et al., 2020). The

*Table 1.* Performance comparison on RGBNT201. The highest and second highest results among all methods are shown in **bold** and in underline respectively. † denotes CLIP-based methods, ∗ denotes ViT-based methods, and the others are CNN-based methods.

| Methods | RGBNT201 | | | |
|---|---|---|---|---|
| | mAP | R-1 | R-5 | R-10 |
| HAMNet (Li et al., 2020) | 27.7 | 26.3 | 41.5 | 51.7 |
| PFNet (Zheng et al., 2021) | 38.5 | 38.9 | 52.0 | 58.4 |
| IEEE (Wang et al., 2022) | 47.5 | 44.4 | 57.1 | 63.6 |
| DENet (Zheng et al., 2023a) | 42.4 | 42.2 | 55.3 | 64.5 |
| LRMM (Wu et al., 2025) | 52.3 | 53.4 | 64.6 | 73.2 |
| UniCat∗ (Crawford et al., 2023) | 57.0 | 55.7 | - | - |
| HTT∗ (Wang et al., 2024b) | 71.1 | 73.4 | 83.1 | 87.3 |
| TOP-ReID∗ (Wang et al., 2024a) | 72.3 | 76.6 | 84.7 | 89.4 |
| EDITOR∗ (Zhang et al., 2024) | 66.5 | 68.3 | 81.1 | 88.2 |
| RSCNet∗ (Yu et al., 2024) | 68.2 | 72.5 | - | - |
| WTSF-ReID∗ (Yu et al., 2025b) | 67.9 | 72.2 | 83.4 | 89.7 |
| DESANet∗ (Dong et al., 2025) | 74.6 | 77.6 | 87.1 | 91.3 |
| MambaPro† (Wang et al., 2025a) | 78.9 | 83.4 | 89.8 | 91.9 |
| DeMo† (Wang et al., 2025b) | 79.0 | 82.3 | 88.8 | 92.0 |
| IDEA† (Wang et al., 2025c) | 80.2 | 82.1 | 90.0 | 93.3 |
| MFRNet† (Feng et al., 2025) | 80.7 | 83.6 | 91.9 | 94.1 |
| Ours† | **83.2** | **87.4** | **92.7** | **94.5** |

*Table 2.* Performance comparison on RGBNT100 and MSVR310. The highest and second highest results among all methods are shown in **bold** and in underline respectively. † denotes CLIP-based methods, ∗ denotes ViT-based methods, and the others are CNN-based methods.

| Methods | RGBNT100 | | MSVR310 | |
|---|---|---|---|---|
| | mAP | R-1 | mAP | R-1 |
| HAMNet (Li et al., 2020) | 74.5 | 93.3 | 27.1 | 42.3 |
| PFNet (Zheng et al., 2021) | 68.1 | 94.1 | 23.5 | 37.4 |
| GAFNet (Guo et al., 2022) | 74.4 | 93.4 | - | - |
| GPFNet (He et al., 2023) | 75.0 | 94.5 | - | - |
| CCNet (Zheng et al., 2023b) | 77.2 | 96.3 | 36.4 | 55.2 |
| LRMM (Wu et al., 2025) | 78.6 | 96.7 | 36.7 | 49.7 |
| GraFT∗ (Yin et al., 2023) | 76.6 | 94.3 | - | - |
| UniCat∗ (Crawford et al., 2023) | 79.4 | 96.2 | - | - |
| PHT∗ (Pan et al., 2023) | 79.9 | 92.7 | - | - |
| HTT∗ (Wang et al., 2024b) | 75.7 | 92.6 | - | - |
| TOP-ReID∗ (Wang et al., 2024a) | 81.2 | 96.4 | 35.9 | 44.6 |
| EDITOR∗ (Zhang et al., 2024) | 82.1 | 96.4 | 39.0 | 49.3 |
| FACENet∗ (Zheng et al., 2025) | 81.5 | 96.9 | 36.2 | 54.1 |
| RSCNet∗ (Yu et al., 2024) | 82.3 | 96.6 | 39.5 | 49.6 |
| WTSF-ReID∗ (Yu et al., 2025b) | 82.2 | 96.5 | 39.2 | 49.1 |
| DESANet∗ (Dong et al., 2025) | 82.1 | 97.4 | 39.2 | 47.8 |
| MambaPro† (Wang et al., 2025a) | 83.9 | 94.7 | 47.0 | 56.5 |
| DeMo† (Wang et al., 2025b) | 86.2 | 97.6 | 49.2 | 59.8 |
| IDEA† (Wang et al., 2025c) | 87.2 | 96.5 | 47.0 | 62.4 |
| MFRNet† (Feng et al., 2025) | **88.2** | 97.4 | 50.6 | 64.8 |
| Ours† | 88.0 | **98.0** | **53.8** | **69.5** |

model is optimized using the Adam optimizer with an initial learning rate of $3.5 \times 10^{-4}$, while a smaller learning rate of $5 \times 10^{-6}$ is applied to fine-tune the visual encoder backbone. Training is performed for 50 epochs, with a batch size of 64 for RGBNT201 and MSVR310, and 128 for RGBNT100. For each identity, 8 pedestrian instances or 16 vehicle instances are randomly sampled. The warm-up phase for the CGSR and CDDO modules lasts 5 epochs on RGBNT201 and RGBNT100, and 10 epochs on MSVR310, where the CGSR weighting coefficient $d$ is fixed at 0.2 during warm-up. All experiments are conducted with PyTorch (Paszke et al., 2019) on an NVIDIA GeForce RTX 4090 GPU.

### 4.3. Comparison with State-of-the-Art Methods

**Multi-modal Person ReID.** Table 1 reports the comparison results between CCL and other state-of-the-art methods on the RGBNT201 dataset. As shown, leveraging complementary semantic information from generated texts, IDEA achieves strong performance. Building upon this, CCL further enhances robustness by dynamically reweighting image–text pairs according to their correspondence degree, effectively suppressing the influence of weak-correspondence pairs. Consequently, CCL surpasses IDEA by 3.0% in mAP and 5.3% in Rank-1 accuracy, and achieves new state-of-the-art performance of 83.2% and 87.4%, respectively, fully unlocking the discriminative potential of strong-correspondence pairs.

**Multi-modal Vehicle ReID.** As shown in Table 2, CCL also demonstrates superior performance on the vehicle re-

identification task. On the RGBNT100 dataset, CCL surpasses the IDEA method which also leverages textual information by 0.8% in mAP, reaching 88.0%. Even on the more challenging MSVR310 dataset, CCL achieves remarkable improvements of 3.2% and 4.7% over the MFRNet method in mAP and Rank-1 accuracy, respectively, fully verifying its reliability and generalization capability in the multi-modal ReID domain.

**Multi-modal Object ReID with Missing Modalities.** To evaluate the robustness of CCL under missing-modality conditions, we conducted comprehensive experiments on the RGBNT201 dataset and compared it with 8 representative methods. As shown in Table 3, the proposed CCL achieves remarkable superiority across almost all missing-modality settings, with average performances of 52.8% and 52.7%. This corresponds to substantial gains of 2.1% and 1.9% over the DeMo baseline, demonstrating the robustness of our correspondence cognitive learning framework under incomplete modality conditions. These results demonstrate that by explicitly modeling the correspondence degree of different pairs and dynamically adjusting the optimization focus, CCL effectively mitigates the uncertainty caused by incomplete modalities, exhibiting stronger robustness and adaptability under missing-modality scenarios.

*Table 3.* **Performance comparison on RGBNT201 of different missing-modality settings.** "M(X)" means missing the X image modality. The highest and second highest results among all methods are shown in **bold** and in underline respectively.

| Type | Methods | M(RGB) | | M(NIR) | | M(TIR) | | M(RGB+NIR) | | M(RGB+TIR) | | M(NIR+TIR) | | Average | |
|---|---|---|---|---|---|---|---|---|---|---|---|---|---|---|---|
| | | mAP | R-1 | mAP | R-1 | mAP | R-1 | mAP | R-1 | mAP | R-1 | mAP | R-1 | mAP | R-1 |
| *Single* | MUDeep (Qian et al., 2017) | 19.2 | 16.4 | 20.0 | 17.2 | 18.4 | 14.2 | 13.7 | 11.8 | 11.5 | 6.5 | 12.7 | 8.5 | 15.9 | 12.9 |
| | HACNN (Li et al., 2018) | 12.5 | 11.1 | 20.5 | 19.4 | 16.7 | 13.3 | 9.2 | 6.2 | 6.3 | 2.6 | 14.8 | 12.0 | 13.3 | 10.7 |
| | MLFN (Chang et al., 2018) | 20.2 | 18.9 | 21.1 | 19.7 | 17.1 | 13.2 | 11.3 | 12.1 | 8.3 | 3.5 | 13.1 | 9.1 | 15.6 | 12.4 |
| | PCB(Sun et al., 2018) | 23.6 | 24.2 | 24.4 | 25.1 | 19.9 | 14.7 | 20.6 | 23.6 | 11.0 | 6.8 | 18.6 | 14.4 | 19.7 | 18.1 |
| | OSNet (Zhou et al., 2019) | 19.8 | 17.3 | 21.0 | 19.0 | 18.7 | 14.6 | 12.3 | 10.9 | 9.4 | 5.4 | 13.0 | 10.2 | 15.7 | 12.9 |
| *Multi* | TOP-ReID (Wang et al., 2024a) | 54.4 | 57.5 | 64.3 | 67.6 | 51.9 | 54.5 | 35.3 | 35.4 | 26.2 | 26.0 | 34.1 | 31.7 | 44.4 | 45.4 |
| | DeMo (Wang et al., 2025b) | 63.3 | 65.3 | 72.6 | 75.7 | 56.2 | 54.1 | **45.6** | **46.5** | 26.3 | 24.9 | 40.3 | 38.5 | 50.7 | 50.8 |
| | MFRNet (Feng et al., 2025) | 64.7 | 65.2 | 72.3 | 76.1 | 51.6 | 49.5 | 41.4 | 43.4 | 27.3 | 27.9 | 37.2 | 35.6 | 49.1 | 49.6 |
| | **Ours** | **66.4** | **67.1** | **75.9** | **80.9** | **58.1** | **56.3** | 43.1 | 40.2 | **30.9** | **31.3** | **42.1** | **40.6** | **52.8** | **52.7** |

## 4.4. Ablation Studies

To investigate the effectiveness of each component, we conduct ablation studies on the RGBNT201 and MSVR310 datasets. As shown in Table 4, we present the ablation results of 6 model variants: (1) Baseline: Baseline of the DeMo method; (2) CCL-1 represents the removal of the CDDO module; (3) CCL-2 represents the removal of the CGSR module; (4) CCL-3 represents the removal of the Hierarchical Decoupling and Attention-Triggered MoE modules from DeMo; (5) CCL-4 represents the removal of the warm-up phase; (6) The full CCL model. It is evident that both CGSR and CDDO significantly improve the baseline performance, and the introduction of the warm-up stage further stabilizes the training process, leading to the best overall results, with mAP values of 83.2% and 53.8% achieved on the RGBNT201 and MSVR310 datasets, respectively.

As for the complexity analysis, although the proposed CGSR module incorporates textual data, it introduces only a minor increase in learnable parameters (less than 15 MB). In contrast, the CDDO module serves as a dynamic learning strategy and does not introduce any additional parameters. Meanwhile, the increase in FLOPs remains small compared with the baseline, demonstrating the effectiveness and efficiency of our method. Notably, CCL-2 represents the variant without the CGSR module, where CDDO is directly applied to the DeMo baseline without introducing textual semantic refinement. Even under this setting, CCL-2 still improves the baseline performance on RGBNT201, demonstrating that the proposed cognitive-driven dynamic optimization itself effectively enhances training robustness by progressively emphasizing reliable image-text pairs.

## 4.5. Visualization Analysis

**Cosine Similarity Distributions.** To verify whether CGSR and CDDO enable the model to learn more discriminative features, Figure 3 illustrates the cosine-similarity distributions of test features. The results show that after injecting CGSR into the baseline, the mean similarity gap between positive and negative samples widens from 0.2585 to 0.2679.

*Table 4.* **Performance comparison with different modules.**

| Type | RGBNT201 | | MSVR310 | | Params | FLOPs |
|---|---|---|---|---|---|---|
| | mAP | R-1 | mAP | R-1 | M | G |
| Baseline | 73.1 | 76.7 | 49.3 | 65.5 | 98.79 | 35.10 |
| CCL-1 | 79.0 | 81.7 | 50.6 | 66.2 | 103.79 | 46.16 |
| CCL-2 | 74.6 | 77.3 | 49.4 | 65.8 | 98.79 | 35.10 |
| CCL-3 | 82.3 | 85.9 | 43.2 | 60.2 | 92.98 | 34.28 |
| CCL-4 | 76.7 | 80.3 | 50.9 | 67.5 | 103.79 | 46.16 |
| The full CCL | **83.2** | **87.4** | **53.8** | **69.5** | 103.79 | 46.16 |

Moreover, the addition of CDDO widens the margin to 0.2967. These results demonstrate that CGSR and CDDO work synergistically to significantly enhance the model's discriminative capability.

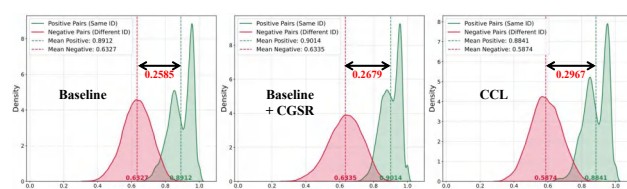

*Figure 3.* Cosine-similarity distribution visualization on the RGBNT201 dataset.

**Multi-modal Feature Distributions.** Figure 4 presents the visualization of discriminative features learned by different variants of CCL. Specifically, compared with Figure 4(a), Figure 4(b) shows that the introduction of CGSR makes the features of the same identity more compact. With the further incorporation of CDDO, Figure 4(c) shows that the model is able to better distinguish instances belonging to different identities. These results demonstrate the effectiveness of our CCL in learning discriminative feature representations.

**Correlation Between Training Loss and Correspondence Degree.** To quantitatively validate the core assumption of the proposed CDDO module, we randomly sample 200 image-text pairs from RGBNT201 at the 5th epoch of the warm-up stage and compute their CLIP-based normalized cosine similarity together with the corresponding training

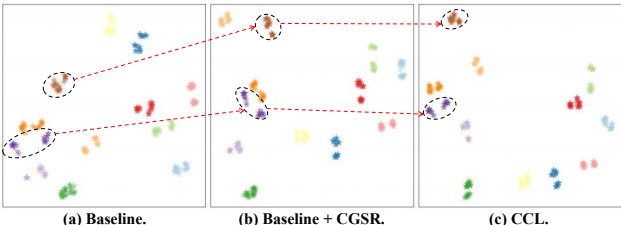

*Figure 4.* t-SNE (Maaten & Hinton, 2008) visualization of feature distributions, with identities coded by distinct colors.

losses. Figure 5 shows a strong negative correlation between the image-text correspondence degree and training loss, achieving a Pearson correlation coefficient of $r = -0.6388$ with $p = 2.49 \times 10^{-24}$. This observation is consistent with the memorization effect of deep neural networks, where strongly-corresponded pairs are learned earlier and tend to produce lower optimization losses. The results provide empirical evidence supporting the rationality of using optimization loss as a proxy for correspondence difficulty estimation.

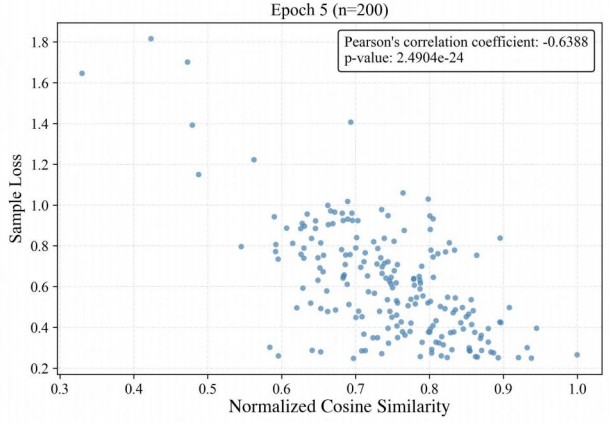

*Figure 5.* Correlation between training loss and image-text correspondence degree on RGBNT201.

**Visualization of Training Process.** To visually demonstrate how CCL performs progressive learning based on the correspondence degree, we illustrate the learning dynamics across samples with different correspondence degrees in Figure 6. As shown in Figure 6(a), during the early training stage, the model assigns higher weights to easy samples with smaller losses to prioritize stable learning, while reducing the weights of hard samples with larger losses to mitigate their negative influence. Subsequently, Figure 6(b) shows that as training progresses, the model gradually increases the weights of hard samples, allowing them to receive greater learning attention. This phenomenon is also clearly verified in Figure 6(c). Through this process, CCL adaptively adjusts its optimization focus according to the correspon-

dence degrees among samples, thereby achieving a stable and cognitive learning paradigm from easy to hard.

**Channel Activation Maps.** Figure 7 visualizes the channel activation maps of the baseline and CCL. As illustrated, CCL dynamically shifts its optimization focus from easy to hard by modeling the image–text correspondence degree, accurately attending to key image regions and demonstrating robust superiority even under challenging scenarios.

### 4.6. More Analysis

**Analysis of parameter** $\gamma$**.** Figure 8 illustrates the impact of the $\gamma$ parameter in our proposed CDDO on model performance. We observe that the model achieves the best performance when $\gamma$ is in the range of $[0.3, 0.6]$. A too-large $\gamma$ assigns excessive weight to weak-correspondence pairs at the beginning of training, misleading the model. Conversely, a too-small $\gamma$ results in uniformly low weights for all samples at the start of training, which hinders effective model initialization.

**Correspondence statistics.** Figure 9 presents the statistical distributions of image-text correspondence degrees on different datasets, where the correspondence degree is measured using the normalized cosine similarity between image and text features. The results clearly demonstrate that the generated textual descriptions exhibit varying semantic alignment quality across different samples, thereby further validating the necessity of explicitly modeling correspondence difficulty during optimization.

In addition, we further analyze the robustness of CCL under weak correspondence scenarios. Specifically, we select the bottom 33% samples with the lowest correspondence scores on RGBNT201 and MSVR310 as weak-correspondence subsets. As shown in Table 6, compared with the text-free DeMo baseline on these challenging subsets, the proposed CCL consistently achieves superior retrieval performance, demonstrating its robustness under imperfect image-text correspondence conditions.

**Comparison with different optimization strategies.** To further validate the effectiveness of our cognitive-driven optimization design, we compare CCL with several representative optimization strategies, including classical SPL strategy, focal loss, and inverse-rank weighting. As shown in Table 5, CCL consistently achieves the best performance. Compared with other optimization strategies, the advantage stems from our method's ability to adaptively balance sample selection and optimization emphasis by progressively prioritizing easy, high-correspondence pairs while suppressing unreliable ones at the early stage of training. These results demonstrate that the proposed correspondence-aware optimization strategy is more suitable for multi-modal ReID with varying image-text correspondence degrees.

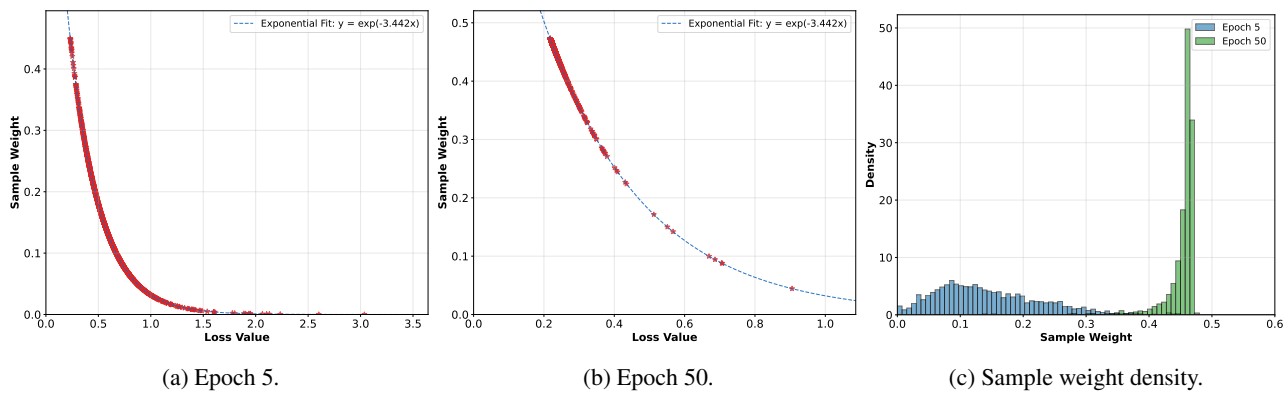

(a) Epoch 5.       (b) Epoch 50.       (c) Sample weight density.

*Figure 6.* Visualization of Training Process.

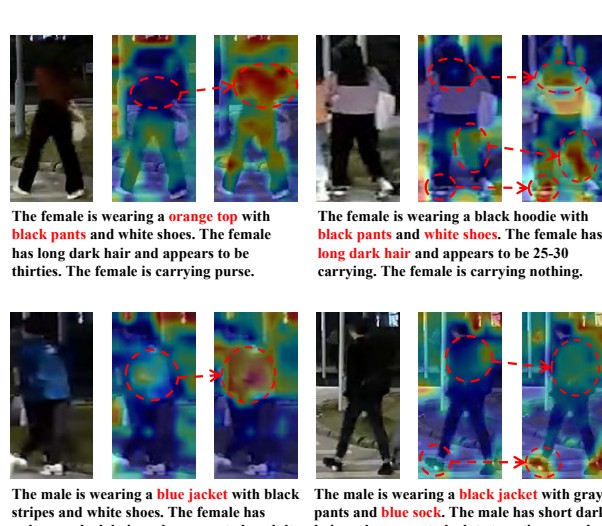

The female is wearing a **orange top** with **black pants** and white shoes. The female has long dark hair and appears to be thirties. The female is carrying purse.

The female is wearing a black hoodie with **black pants** and **white shoes**. The female has **long dark hair** and appears to be 25-30 carrying. The female is carrying nothing.

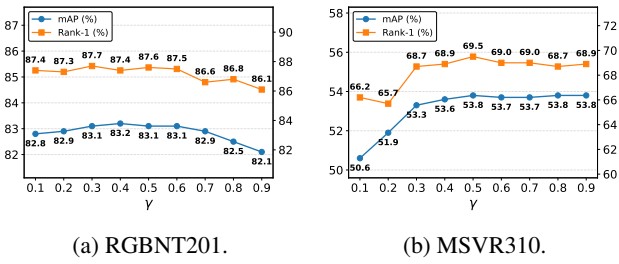

The male is wearing a **blue jacket** with black stripes and white shoes. The female has unknown dark hair and appears to be adult. The female is carrying nothing.

The male is wearing a **black jacket** with gray pants and **blue sock**. The male has short dark hair and appears to be late twenties or early thirties. The male is carrying nothing.

*Figure 7.* Channel Activation Maps of the original image, the baseline, and the CCL model.

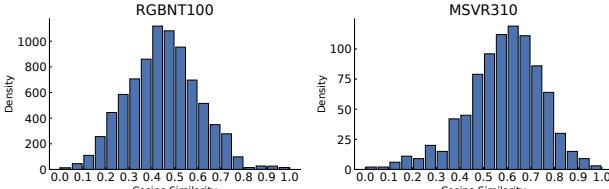

(a) RGBNT201.       (b) MSVR310.

*Figure 8.* Performance comparison with different $\gamma$ on RGBNT201 and MSVR310 datasets.

*Figure 9.* Statistics of image-text correspondence degree on RGBNT100 and MSVR310 datasets.

*Table 5.* Comparison with different optimization strategies.

| Methods | RGBNT201 | | | |
|---|---|---|---|---|
| | mAP | R-1 | R-5 | R-10 |
| Classical SPL | 81.4 | 86.4 | 92.0 | 93.8 |
| Focal Loss | 79.4 | 82.4 | 90.8 | 92.9 |
| Inverse-rank Weighting | 80.1 | 84.6 | 90.3 | 93.7 |
| **CCL** | **83.2** | **87.4** | **92.7** | **94.5** |

*Table 6.* Performance comparison on weak-correspondence subsets selected from RGBNT201 and MSVR310.

| Dataset | Method | mAP | R-1 | R-5 |
|---|---|---|---|---|
| RGBNT201 | DeMo | 60.4 | 61.5 | 75.1 |
| | **CCL** | **62.5** | **63.2** | **76.2** |
| MSVR310 | DeMo | 33.1 | 50.3 | 71.1 |
| | **CCL** | **33.9** | **52.8** | **72.3** |

# 5. Conclusion

In this paper, we propose a new CCL framework for multi-modal object ReID that explicitly models the correspondence degree and facilitates a progressive learning process from easy to hard pairs. CCL consists of two modules, i.e., CGSR and CDDO. Specifically, CGSR dynamically re-fines visual representations using text semantics according to the correspondence difficulty estimated from the previous epoch, thereby enhancing feature alignment under imperfect associations. Additionally, CDDO presents a self-paced weighting mechanism that adaptively adjusts the optimization focus by emphasizing easy pairs at the early stage and gradually integrating harder ones as training evolves. Extensive experiments on three multi-modal object ReID benchmarks demonstrate the superior performance of CCL.

## Acknowledgements

This work was supported in part by the National Natural Science Foundation of China under Grant 62372315, Sichuan Science and Technology Planning Project (Grant No. 2026NSFSC1480 and 2024ZDZX0004), Central Government's Guide to Local Science and Technology Development Fund under Grant 2025ZYDF101, Chengdu Science and Technology Project (Grants No. 2025-YF08-00104-GX and 2025-YF05-00169-SN).

## Impact Statement

This paper presents work whose goal is to advance the field of Machine Learning. There are many potential societal consequences of our work, none of which we feel must be specifically highlighted here.

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

# Appendix

In the following sections, we provide the details about the theoretical rationality of our CDDO (Sec. A) and more visualization analysis (Sec. B) of our proposed CCL.

## A. Theoretical Justification

To further explain the theoretical rationality of our Cognitive-Driven Dynamic Optimization (CDDO), we analyze the mathematical properties of the exponential self-paced weighting function:

$$w(l) = e^{-l/\rho}, \quad \rho > 0,$$

where $l$ denotes the per-sample loss and $\rho$ is a temperature-like scale parameter. We define the per-sample contribution function as:

$$g(l) = w(l)\, l = l\, e^{-l/\rho}.$$

The behavior of $g(l)$ determines how different samples participate in optimization. Below, we present three propositions showing that this design naturally realizes a self-paced curriculum, implicitly models the correspondence degree between image–text pairs, and guarantees robustness.

**Proposition 1. (Self-paced learning property)** The contribution function $g(l) = le^{-l/\rho}$ is unimodal and strictly increasing when $l < \rho$, reaching its maximum at $l = \rho$ with $g_{\max} = \rho e^{-1}$, and strictly decreasing when $l > \rho$:

$$g'(l) = e^{-l/\rho}\left(1 - \frac{l}{\rho}\right). \tag{12}$$

Hence, samples with smaller losses (i.e., easier samples) contribute more to the objective, while extremely hard or noisy samples with large losses are automatically down-weighted.

*Conclusion.* This unimodal shape induces a continuous and smooth *self-paced curriculum*: the model first focuses on low-loss (easy) samples and gradually incorporates harder ones as their losses decrease during training, without requiring explicit sample ranking or thresholds.

**Proposition 2. (Correspondence-adaptive weighting)** Let each image–text pair have a correspondence degree $c \in [0, 1]$, and assume the expected loss decreases monotonically with $c$, i.e.,

$$\mathbb{E}[l|c_1] < \mathbb{E}[l|c_2], \quad \text{for } c_1 > c_2. \tag{13}$$

Since $w(l) = e^{-l/\rho}$ is strictly decreasing in $l$, it follows that

$$\mathbb{E}[w|c_1] > \mathbb{E}[w|c_2]. \tag{14}$$

*Conclusion.* The exponential weight thus implicitly models the image–text correspondence degree: strong-correspondence pairs (with higher $c$) are assigned larger weights, while weak-correspondence or mismatched pairs receive smaller weights. This correspondence-adaptive weighting reinforces semantically aligned samples and suppresses noisy ones, improving cross-modal matching stability.

**Proposition 3. (Bounded influence and robustness)** The per-sample contribution $g(l) = le^{-l/\rho}$ is globally bounded:

$$0 \le g(l) \le \rho e^{-1}, \quad \forall l \ge 0. \tag{15}$$

The gradient of this term w.r.t. model parameters $\theta$ satisfies:

$$\nabla_\theta g(l(\theta)) = e^{-l/\rho}\left(1 - \frac{l}{\rho}\right)\nabla_\theta l(\theta), \tag{16}$$

implying that $\|\nabla_\theta g(l(\theta))\| \to 0$ exponentially as $l \to \infty$.

*Conclusion.* Each sample's contribution to the overall objective is upper-bounded by $\rho e^{-1}$, ensuring that no single outlier can dominate the optimization. Meanwhile, the exponentially decaying gradient suppresses the influence of large-loss samples, thereby guaranteeing optimization-level *robustness* against noisy or weak-correspondence pairs.

# B. More Visualization Analysis of CCL

## B.1. More Visualization for Person ReID

**Visualization of Channel Activation Maps on the Person ReID Dataset.** Figure 12(a) presents the channel activation maps of various modalities produced by CCL on person ReID datasets. The modalities show different activation behaviors that stem from their respective spectral traits. Notably, the maps are able to pinpoint key semantic areas, including hair, apparel, and accessories, which illustrates the ability of our modules to make effective use of multi-modal cues.

**Visualization of Multi-modal Ranking List with Different Modules in CCL.** In Figure 10, we illustrate the impact of

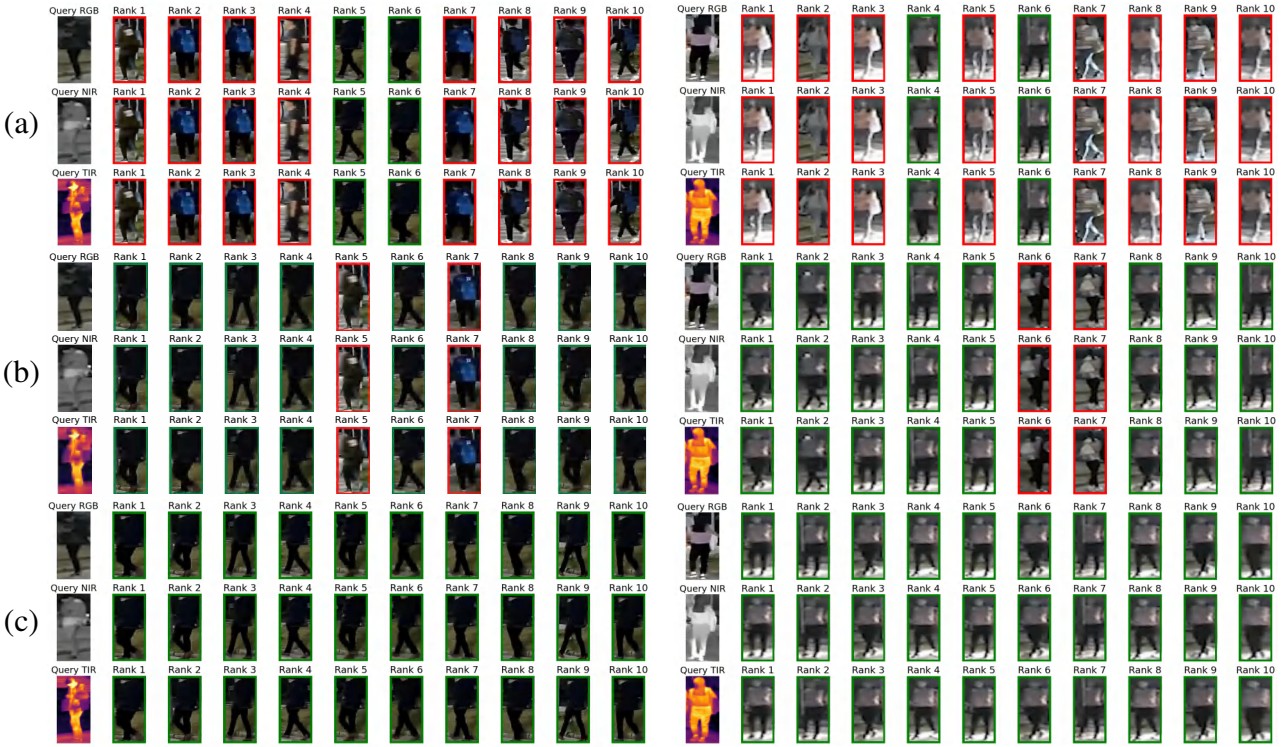

*Figure 10.* Ranking list comparison with different modules on the person ReID dataset RGBNT201. (a) Baseline. (b) Baseline + CGSR. (c) CCL. The green box indicates the correct match, while the red box indicates the incorrect match.

different components on multi-modal ranking results. Specifically, Figure 10(a) shows that the baseline model performs poorly in matching due to the lack of guidance from text information. In Figure 10(b), leveraging text information to enhance visual features significantly reduces incorrect matches. Furthermore, Figure 10(c) achieves the best performance by learning correspondence degree pairs from easy to hard, thereby avoiding the misleading influence of weak-correspondence pairs. These results demonstrate the effectiveness of each module in CCL.

**Visualization of Multi-modal Ranking List Comparison with IDEA.** Figure 11 presents a comparison of multi-modal matching results between CCL and IDEA. It can be seen that CCL achieves significant improvements over IDEA, particularly on challenging samples where images are blurry or targets are unclear. This demonstrates that our proposed learning strategy, which progressively learns based on different correspondence degrees between pairs, can effectively prevent the model from being misled by weak-correspondence pairs, enabling the model to achieve robust and superior performance.

## B.2. More Visualization for Vehicle ReID

**Visualization of Channel Activation Maps on the Vehicle ReID Dataset.** Figure 12(b) presents the channel activation maps of different modalities in the CCL framework on the vehicle ReID dataset. Each modality focuses on distinct semantic regions, such as the vehicle's main structure, edge areas, and the license plate. These activation results highlight key and discriminative local details, further demonstrating that our CCL method effectively leverages the visual information described in the text.

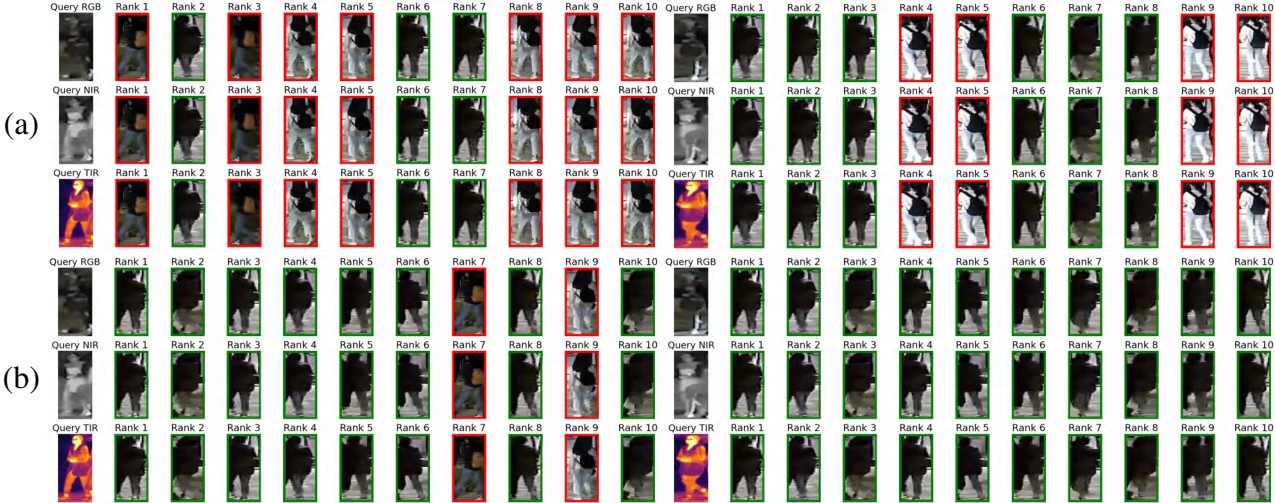

*Figure 11.* Multi-modal ranking list comparison with IDEA on the person ReID dataset RGBNT201. (a) IDEA. (b) CCL.

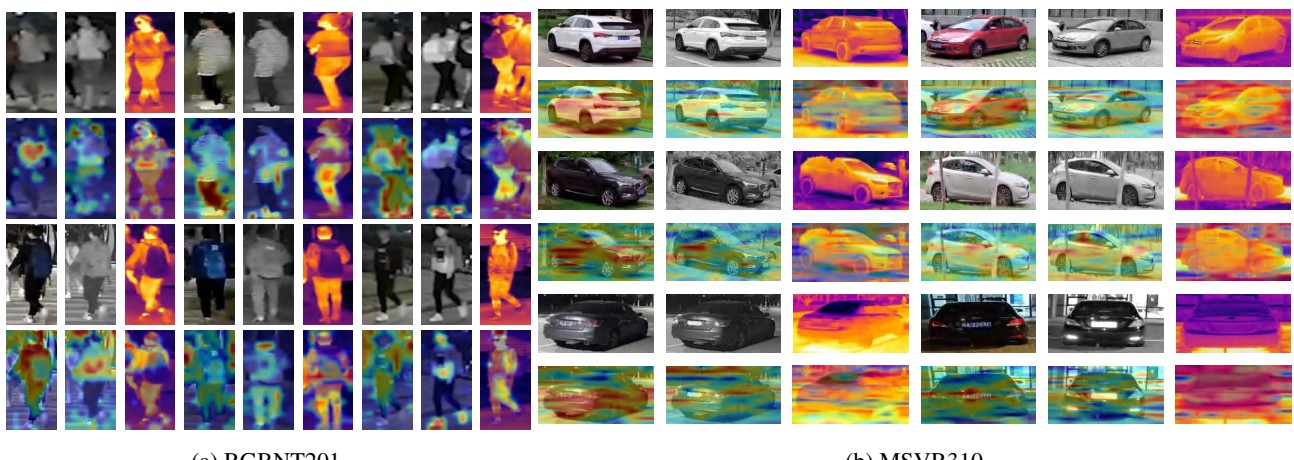

(a) RGBNT201.                                                    (b) MSVR310.

*Figure 12.* Visualization of channel activation maps of different modalities on the RGBNT201 and MSVR310 datasets.

**Visualization of Multi-modal Ranking List with Different Modules in CCL.** In the more challenging vehicle ReID task, Figure 13 illustrates the impact of different components on the ranking results of the MSVR310 dataset. Specifically, Figure 13(a) shows that the baseline model performs poorly without guidance from text information. By incorporating text information (Figure 13(b)), the model can enhance visual features, particularly vehicle contours and license plate details, significantly reducing incorrect matches. Furthermore, Figure 13(c) achieves the best performance by learning correspondence degree pairs in an easy-to-hard manner, effectively avoiding the misleading influence of weak-correspondence pairs. These results clearly demonstrate the effectiveness of the proposed modules.

**Visualization of Multi-modal Ranking List Comparison with IDEA.** Figure 14 and Figure 15 present comparisons of multi-modal matching results between CCL and IDEA on the more challenging vehicle ReID datasets. It is evident that CCL achieves significant improvements over IDEA, especially on difficult samples where vehicle edges are unclear and license plates are blurry. This clearly demonstrates that our proposed progressive learning strategy, based on different correspondence degrees between pairs, effectively mitigates the interference of weak-correspondence pairs, allowing the model to maintain robust and superior performance even on challenging samples.

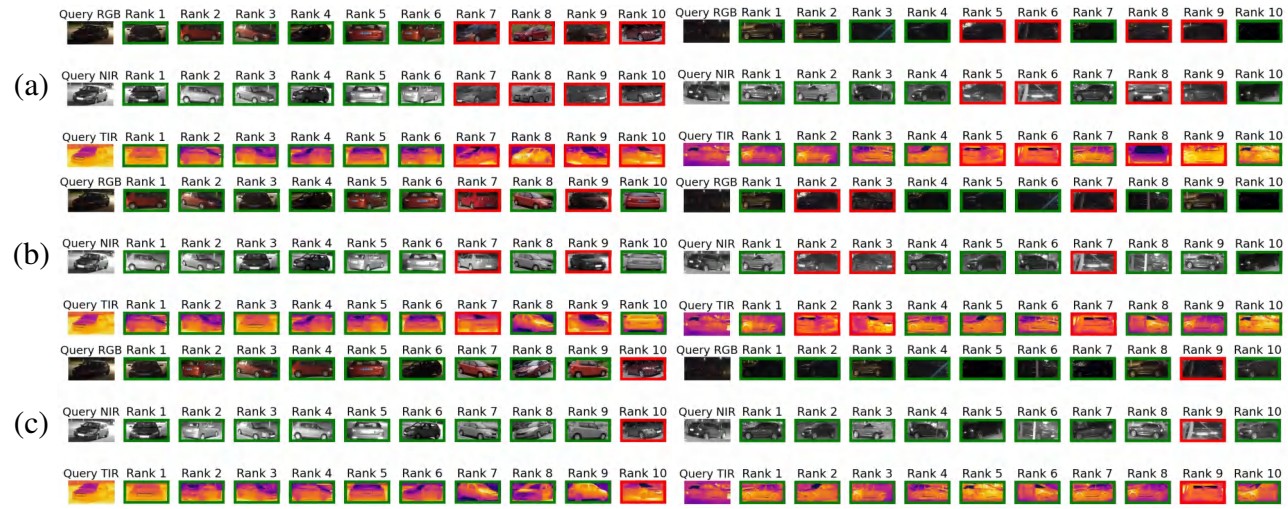

*Figure 13.* Ranking list comparison with different modules on the vehicle ReID dataset MSVR310. (a) Baseline. (b) Baseline + CGSR. (c) CCL. The green box indicates the correct match, while the red box indicates the incorrect match.

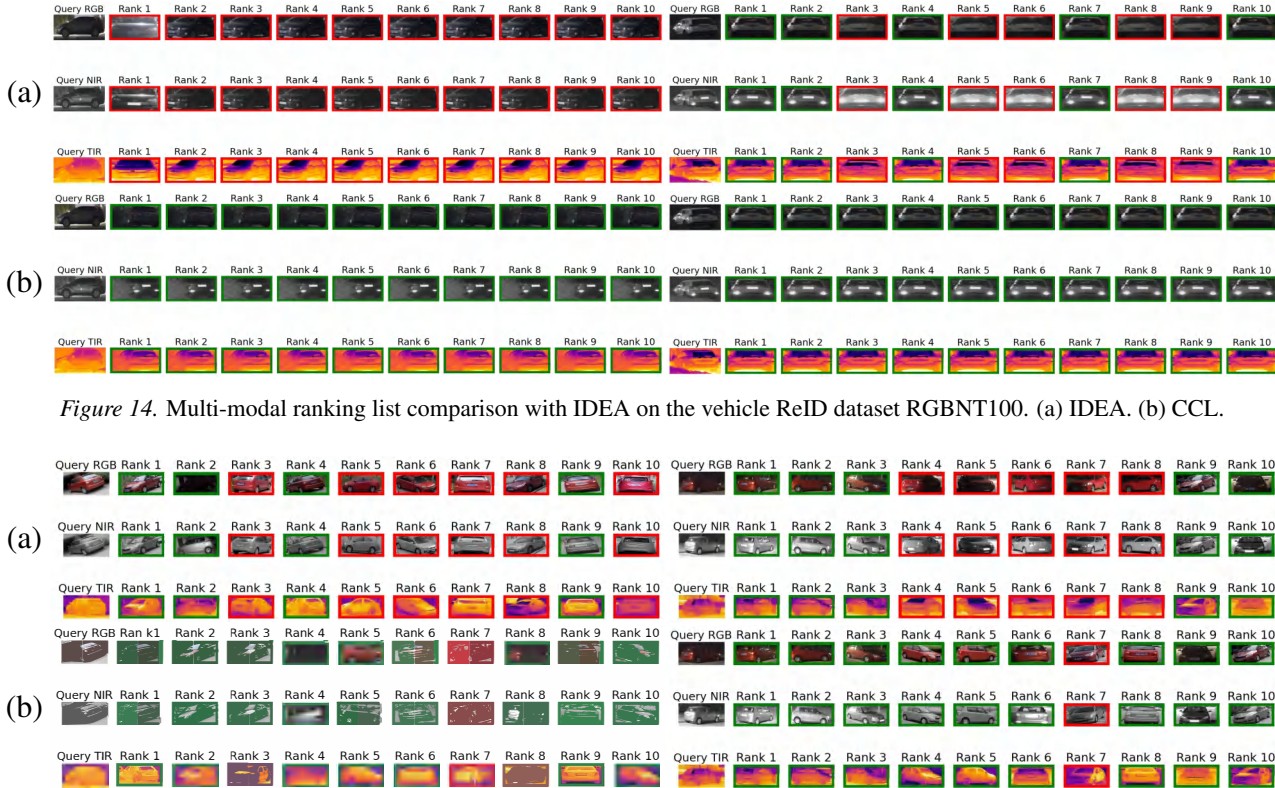

*Figure 14.* Multi-modal ranking list comparison with IDEA on the vehicle ReID dataset RGBNT100. (a) IDEA. (b) CCL.

*Figure 15.* Multi-modal ranking list comparison with IDEA on the vehicle ReID dataset MSVR310. (a) IDEA. (b) CCL.

## C. Case Study of Image-Text Correspondence

Figure 16 presents a comparison between large-loss and low-loss sample pairs. As can be observed, samples with a strong image–text correspondence lead to lower training loss and are assigned higher learning weights by our CCL method. In contrast, samples with weaker correspondence incur larger loss and receive lower learning weights. This phenomenon demonstrates the necessity of the easy-to-hard learning strategy adopted in our CCL framework.

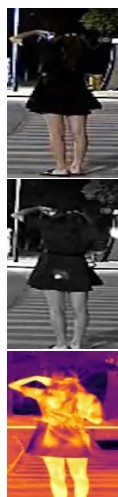

**RGB**：The female is wearing a black dress with white polka dots and sandal shoes. The female has unknown unknown hair and appears to be young adult. The female is carrying nothing.

**NIR**：The female is wearing a black unknown pattern with black potentially textured and vague barefoot. The female has long dark hair and appears to be cannot determine. The female is carrying nothing.

**TIR**：The female is wearing a orange top with black skirt and white flip-flops. The female has long and straight brown hair and appears to be late teens to early twenties. The female is carrying nothing.

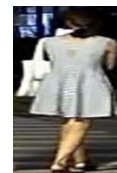
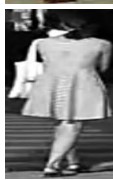
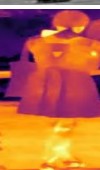

**RGB**：The female is wearing a vague gray dress with white polka dots with vague black sandals and black shoes. The female has ponytail unknown hair and appears to be 20-30. The female is carrying white handbag.

**NIR**：The female is wearing a white dress with black stripes and dots and sandal shoes. The female has unknown tied back hair and appears to be middle-aged. The female is carrying handbag.

**TIR**：The female is wearing a orange top with blue skirt and yellow sandals. The female has long and straight brown hair and appears to be middle-aged. The female is carrying handbag.

Low Loss: 0.22
High Weight: 0.47

Low Loss: 0.22
High Weight: 0.47

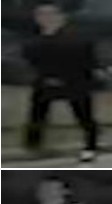

**RGB**：The female is wearing a black hoodie with white pants and black sneakers. The female has short dark brown hair and appears to be young adult. The female is carrying unknown.

**NIR**：The female is wearing a vague black jacket with vague dark pants and vague shoes. The female has short brown hair and appears to be young adult. The female is carrying nothing.

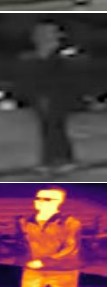

**TIR**：The male is wearing a black jacket with white striped shirt and brown shoes. The male has short dark hair and appears to be middle-aged. The male is carrying nothing.

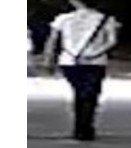

**RGB**：The female is wearing a white t-shirt with black pants and sandal shoes. The female has unknown bun hair and appears to be young adult. The female is carrying nothing.

**NIR**：The male is wearing a white t-shirt with black pants and vague shoes. The male has short dark brown hair and appears to be late teens to early twenties. The male is carrying nothing.

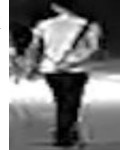

**TIR**：The male is wearing a orange shirt with blue pants and brown shoes. The male has unknown dark hair and appears to be middle-aged. The male is carrying nothing.

Large Loss: 0.95
Low Weight: 0.04

Large Loss: 1.04
Low Weight: 0.03

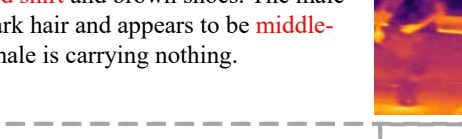

*Figure 16.* Case Study of Image-Text Correspondence.

