# OpenReview forum: "Correspondence Cognitive Learning for Multi-Modal Object Re-Identification"
_ICML.cc/2026/Conference — ICML 2026 regular_

### Official Review · Reviewer_nk1d · 2026-02-20

**Soundness:** 3
**Presentation:** 3
**Significance:** 3
**Originality:** 3
**Overall Recommendation:** 4
**Confidence:** 5

**Summary:**

The paper studies multi-modal object re-identification with auxiliary text generated by a multimodal LLM, focusing on the problem that existing methods treat all image–text pairs as equally reliable. The authors propose a Correspondence Cognitive Learning (CCL) framework that explicitly models correspondence difficulty and uses it both for feature refinement and optimization. The Correspondence-Guided Semantic Refinement (CGSR) module injects text semantics into visual features via cross-attention, token-wise semantic masks, and a global difficulty-based fusion gate. The Cognitive-Driven Dynamic Optimization (CDDO) module applies a self-paced exponential weighting of per-sample losses to emphasize easier, strong-correspondence pairs early in training and progressively include harder ones. Experiments on three multi-modal ReID benchmarks (RGBNT201, RGBNT100, MSVR310) show consistent improvements over competitive baselines, including recent text-augmented approaches.

**Compliance With Llm Reviewing Policy:**

Affirmed.

**Final Justification:**

I thank the authors for their detailed response. The newly added experimental results effectively demonstrate the negative correlation between the training loss and the image-text correspondence, which successfully resolves one of my primary concerns.

**Key Questions For Authors:**

$\textbf{How correspondence-specific is CDDO?}$

Have you tried applying CDDO (with the same exponential weighting and regularizer) to the DeMo baseline \emph{without} any textual information or CGSR? If so, how much of the improvements in Tables~1--2 can be attributed purely to generic robust/self-paced training, versus text-aware refinement? Reporting this ablation (DeMo + CDDO only) would substantially clarify the contribution of correspondence modeling and could materially affect my evaluation.

$\textbf{Correlation between correspondence and loss/$d_i$.}$

Do you have empirical evidence, beyond the anecdotal Fig.~14, that early-stage loss or $d_i$ correlates with actual image--text correspondence quality? For instance, could you approximate correspondence using CLIP image--text similarity, or manually annotate a small subset of pairs, and then report correlation coefficients (e.g., Spearman/Pearson) between correspondence scores and loss/$d_i$? Demonstrating a clear positive correlation would make the ``cognitive correspondence'' narrative considerably more convincing.

$\textbf{Implementation details of $\Omega$, $d_i$, and $\rho$.}$

Could you precisely specify how $\Omega$ and $d_i$ are computed and updated across epochs (e.g., sum vs.\ average of losses, smoothing/EMA, and whether $d_i$ is frozen after warm-up or updated periodically)? Additionally, how often is $\rho$ recalculated, and is it computed per batch (using the batch $\gamma$-quantile) or from a global pool of losses? These details are crucial for reproducibility.

$\textbf{Behavior on RGBNT100.}$

In Table 2, your method attains slightly lower mAP than MFRNet on RGBNT100, despite higher Rank-1. Do you have multi-seed results with variance estimates, or a plausible explanation (e.g., overfitting, data imbalance, or dataset-specific characteristics) for this behavior? Clarifying this point would affect whether the method should be interpreted as consistently superior across datasets.

**Limitations:**

$\textbf{Fig.1(b): correspondence degree is undefined and text may hurt under weak correspondence.}$

Figure~1(b) claims that ``key attributes visibly deviate from the text'' for weak-correspondence pairs, yet the paper does not define how the correspondence degree is quantified or determined beyond illustrative examples. Please provide a reproducible correspondence metric / labeling protocol and report dataset-level statistics. More importantly, please empirically verify that the injected text semantics \emph{improves rather than harms} feature learning under weak correspondence, e.g., by reporting performance stratified by correspondence bins and conducting a caption-shuffling (mismatched text) stress test.


$\textbf{Cognitive Difficulty Estimation: loss-based proxy may reflect generic ReID hardness, not correspondence.}$

Cognitive Difficulty Estimation uses the accumulated CE+Triplet losses during warm-up as a proxy for ``correspondence difficulty'' (Eq.~1), which appears to capture intrinsic visual/ReID hardness rather than image--text correspondence reliability. Please justify this proxy via correlation analysis against a direct correspondence measure (e.g., image--text matching scores or human-rated correspondence), and compare with alternative difficulty estimators derived from cross-modal alignment signals.


$\textbf{Ambiguous definition of easy samples/pairs across the paper.}$

The definition of ``easy samples/pairs'' is ambiguous across the introduction and methods: it is sometimes tied to strong correspondence, while CDDO ties it to low losses via a $\gamma$-quantile threshold (Eqs.9--10), and CGSR uses a Min--Max normalized difficulty $d_i$. Please provide a unified, explicit definition and a consistent selection criterion.
\end{enumerate}

**Strengths And Weaknesses:**

$\textbf{1. Strengths}$

(1) On RGBNT201 (Table 1), the method improves over a strong text-augmented baseline IDEA by +3.0 mAP and +5.3 Rank-1, and also beats the very recent DeMo and MFRNet baselines.
On RGBNT100 and MSVR310 (Table 2), it matches or exceeds the best existing results; the +3.2 mAP and +4.7 Rank-1 gains over MFRNet on MSVR310 are particularly notable given that these are already competitive models.

(2) Robustness in missing-modality scenarios. Table 3 evaluates a rich set of missing-modality combinations on RGBNT201. CCL improves average mAP/R1 by about 2 points over DeMo, which directly supports the claim that correspondence-aware learning helps under degraded conditions.

$\textbf{2. Weaknesses}$

(1) $\textbf{Limited disentangling of correspondence-aware effects vs.\ generic robust training.}$

The paper heavily emphasizes correspondence modeling, yet CDDO largely resembles a generic self-paced reweighting scheme driven by training loss. There is no comparison against classical self-paced learning (SPL) or standard robust objectives (e.g., focal loss, loss-based sample reweighting, inverse-rank weighting), making it plausible that a substantial portion of the gains stems from generic robustness rather than image--text correspondence per se. Without these baselines, the conceptual framing as cognitive correspondence learning appears somewhat overstated.

(2) $\textbf{Weak empirical support that loss is a reliable proxy for correspondence.}$

Both difficulty $d_i$ and weights $w_i$ are derived from loss signals, implicitly assuming a monotonic relation between correspondence quality and expected loss. However, beyond a single illustrative example (Fig.14) and qualitative visualizations (e.g., Fig.5), the paper lacks systematic quantitative validation (e.g., correlation between human-labeled correspondence scores and $d_i/w_i$, or calibration curves). This creates a gap between the proposed narrative and the empirical evidence.

(3) $\textbf{Ambiguity in the difficulty estimation procedure.}$

Section 3.1 describes accumulating loss into $\Omega$ and normalizing to obtain $d$, but key implementation details are unclear: are losses summed or averaged across iterations/epochs, is any smoothing (EMA) applied, and are $d_i$ values frozen after warm-up or updated periodically? Since $d_i$ governs CGSR fusion and how textual cues are used throughout training, these details are important for reproducibility.

(4) $\textbf{Insufficient analysis of CGSR internal behavior.}$

Although Fig.2 sketches CA, SA, and the semantic mask, the paper provides limited quantitative insight into $S^m$ (e.g., distribution statistics for strong vs.\ weak correspondence samples, sensitivity to thresholds). It also does not clearly specify how the ``secondary query'' used to construct anchors $s_j^m$ is implemented, which is essential for Eqs.(3)--(4). As written, the mask may be perceived as a black-box scoring module with limited interpretability.

(5) $\textbf{Modest/ambiguous improvements on RGBNT100.}$

In Table 2, CCL achieves slightly lower mAP than MFRNet on RGBNT100 (88.0 vs.\ 88.2) despite a higher Rank-1. Without error bars or repeated runs, it is unclear whether CCL offers a statistically meaningful improvement on this dataset. While the claimed gain over IDEA (e.g., +0.8 mAP) is fair, the broader impression that CCL consistently surpasses the strongest prior method is not strictly supported.

(6) $\textbf{Missing run-to-run variability and significance testing.}$

Tables 1--4 report single-run numbers only. For large improvements this is less concerning, but for small deltas (e.g., +0.8 mAP on RGBNT100, some missing-modality settings in Table~3), variance could change conclusions. Reporting mean$\pm$std over multiple seeds and, where appropriate, significance tests would strengthen the empirical claims.

(7) $\textbf{Narrow experimental scope and incomplete benchmark coverage.}$

All experiments are restricted to the RGBNT-family benchmarks (RGB/NIR/TIR), yet the method is positioned as broadly applicable to multi-modal object Re-ID. Moreover, commonly used RGB-thermal video benchmarks such as MSVR310 are not included. Evaluating on at least one additional dataset outside RGBNT (or a synthetic cross-domain setting) would better support generality.

(8) $\textbf{Missing closely related literature.}$

The related work omits highly relevant recent methods on text-modulated multi-modal Re-ID, e.g., $\textbf{NEXT: Multi-Grained Mixture of Experts via Text-Modulation for Multi-Modal Object Re-ID}$, which should be discussed and contrasted conceptually and experimentally where feasible.

---

> ### Author Rebuttal · Authors · 2026-03-31
>
> **Reply to Weakness (1)**:
> Thanks. We conducted additional experiments comparing CCL with several robust baselines on RGBNT201 dataset.
> The experimental results (see https://anonymous.4open.science/r/CCL_Figure-1662/nk1d_weakness_1.png) consistently demonstrate that CCL outperforms all generic robust baselines, validating that our CDDO module captures unique image-text correspondence beyond the reach of generic robustness.
>
> **Reply to Weakness (2), Question (2), Limitations (2)**:
> Thanks. The motivation behind CDDO is grounded in the "memorization effect" of Deep Neural Networks  (Please see lines 231-241 in Section 3.2).
> To quantitatively validate our core hypothesis, we randomly sampled 200 pairs at the 5th epoch of the warm-up phase and computed their CLIP-based normalized cosine similarity scores alongside training losses. The results reveal a strong negative correlation between early loss values and objective image-text similarity (Pearson r=−0.6388 , p=2.4904e-24, see https://anonymous.4open.science/r/CCL_Figure-1662/RGBNT201_Correlation.jpg).
>
>
> **Reply to Weakness (3) and Question (3)**:
> Thanks. For $\rho$, it is computed as the $\gamma$-quantile of the global loss pool from the final epoch of warm-up, and $\rho$ is frozen after this computation. After warm-up, the weight $w$ is calculated based on the instantaneous loss of the current epoch. The global loss pool is normalized to obtain the global difficulty pool $\Omega$.
> Furthermore, starting from the final epoch of warm-up, $d_i$ is derived from the globally accumulated difficulty pool $\Omega$ of the previous epoch. No smoothing/EMA is applied to this loss, and the global difficulty pool $\Omega$ is updated every epoch.
>
> **Reply to Weakness (4)**:
> Thanks. Regarding the spatially-grounded semantic anchor $s_j^m$ obtained via the "secondary query".specifically, we employ $\mathbf{F}_{mix}^m$ as the query to perform a cross-attention operation over the textual features.
>
> **Reply to Weakness (5)(6) and Question (4)**:
> Thanks. First, it is important to clarify that the results of the comparative methods reported in the table are all taken directly from their original papers. This is not only a common practice, but also helps avoid unfair comparisons caused by potential performance degradation in reproduced results.
> Nevertheless, in the course of this work, we actually conducted experiments with six different random seeds for DeMo, IDEA, MFRNet, and our CCL method (see https://anonymous.4open.science/r/CCL_Figure-1662/nk1d_weakness_5.png). Here, we report the mean$\pm$std of these four methods on the RGBNT100 dataset.
>
> **Reply to Weakness (7) **:
> In fact, MSVR310 is a primary benchmark in our paper, with extensive results already provided: Main Results (Table 2), Ablation Study (Table 4), Parameter Analysis (Fig. 6(c)), Visualization of channel activation maps (Fig. 10) and Ranking list comparison (Fig 11, 13).
>
> **Reply to Weakness (8) **:
> Thanks. The reasons for not including NEXT in our comparison are **Lack of publicly available source code** and **Different text description data and prompt strategies from our work**.
>
> **Reply to Question (1)**:
> Thanks. In fact, we have conducted these ablation experiments. As shown in Table 4, CCL-2 represents the removal of the CGSR module, i.e., applying CDDO (with the same exponential weighting and regularizer) to the DeMo baseline \emph{without} any textual information. For the RGBNT201 dataset, the CCL-2 variant achieves a 1.5% improvement in mAP over the baseline (DeMo).
>
>
> **Reply to Limitations (1)**: Thanks. Following your suggestion, we use the normalized cosine similarity between images and texts to explicitly define the correspondence degree and report its statistics for different datasets
> (https://anonymous.4open.science/r/CCL_Figure-1662/RGBNT201_statistics.png for RGBNT201, https://anonymous.4open.science/r/CCL_Figure-1662/RGBNT100_statistics.png for RGBNT100, and https://anonymous.4open.science/r/CCL_Figure-1662/MSVR310_statistics.png for MSVR310).
>
> We then select the bottom 33% of samples with the lowest correspondence degree on RGBNT201 and MSVR310 as subsets representing weak correspondence, and compare the retrieval performance of the proposed CCL with DeMo (without using text) on these subsets.  (see https://anonymous.4open.science/r/CCL_Figure-1662/nk1d_limitation_1.png)
>
> Furthermore, for stress-testing under shuffled (mismatched) text conditions, please see Reply to Weakness (4) of Review #du3N.
>
>
> **Reply to Limitations (3)**
>  Strong-correspondence pairs (“Simple Samples”) have lower training loss, while weak-correspondence pairs (“Hard Samples”) have higher loss. Both CGSR and CDDO use the accumulated training loss $\mathcal{L}_i$ to indicate difficulty. The difference in formulation reflects their roles: CGSR applies Min-Max normalization ($d_i \in [0,1]$) for per-sample gating in feature fusion, whereas CDDO uses $p$-quantile thresholds for batch-level weight reassignment.

---

> > ### Author Rebuttal · Reviewer_nk1d · 2026-04-05
> >
> > My concerns have been adequately addressed.

---

### Official Review · Reviewer_oL7b · 2026-03-06

**Soundness:** 4
**Presentation:** 4
**Significance:** 4
**Originality:** 4
**Overall Recommendation:** 5
**Confidence:** 5

**Summary:**

Existing multi-modal object Re-Identification (ReID) approaches usually overlook the varying correspondence degrees between visual and textual modalities, which limits their ability to learn discriminative associations and hinders effective optimization. To address this limitation, this paper proposes a Correspondence Cognitive Learning (CCL) framework that explicitly models the correspondence degree and facilitates a progressive learning process from easy to hard pairs. Extensive experiments on three multi-modal object ReID benchmarks demonstrate the superior performance of the proposed CCL method.

**Compliance With Llm Reviewing Policy:**

Affirmed.

**Final Justification:**

The authors have adequately addressed my questions and further provided analyses and experimental evidence that better support the core motivation of the paper. In particular, rather than treating generated text as a uniform supervision signal, the paper explicitly models correspondence difficulty between image-text pairs and incorporates it into both feature refinement and optimization. I consider this to be a key novelty of the work, and it clearly distinguishes the proposed method from prior multi-modal object ReID approaches.

I have also read the other reviewers’ comments and the authors’ responses. Overall, the main concerns have been handled appropriately, and I do not see any remaining issue that would materially weaken the paper’s contribution or conclusions. Therefore, I maintain my original assessment and continue to recommend acceptance.

**Key Questions For Authors:**

Q1: What advantages does the Correspondence-Guided Fusion in CGSR have over the conventional Cross-Attention Mechanism?

Q2: How should the sparsity shown in Figures 5(a) and 5(b) be interpreted? What is the relationship between this experimental phenomenon and the correspondence degrees?

**Limitations:**

Although the authors have designed an effective CCL method, the presentation of the experimental results is somewhat rough. The categorization of comparison methods in the tables and the readability of all figures need to be further improved.

**Strengths And Weaknesses:**

**Strengths**

1) The proposed CCL method addresses the issue that existing multi-modal object ReID approaches overlook the correspondence degrees between visual and textual modalities, and enables a progressive learning process from easy to hard pairs.
2) By linking the correspondence degrees with the training loss, CDDO presents a self-paced weighting mechanism that adaptively adjusts the optimization focus.
3) The visualization results in the paper are rich and intuitively demonstrate the effectiveness of the proposed CGSR and CDDO.
4) The paper is clearly written and well organized, effectively explaining the motivation and details of the proposed CCL method.

**Weaknesses**

1) The categorization of the comparison methods in Tables 1 and 2 is not clear.
2) The text in the figures is too small, which makes it difficult to read the details.
3) The paper does not seem to clearly explain the relationship between self-paced learning and cognitive learning.

---

> ### Author Rebuttal · Authors · 2026-03-31
>
> **Reply to Weaknesses (1)**: Thanks for your valuable question. In the final version of the paper, we will categorize the methods based on the backbone architecture employed by each method.
>
> **Reply to Weaknesses (2)**: Thanks for your valuable question. We will increase the font size in the figures in subsequent versions of the paper to improve readability.
>
> **Reply to Weaknesses (3)**: Thanks for your valuable question. While CDDO formally utilizes training loss as a guide, it is fundamentally distinct from generic Self-Paced Learning (SPL): our experiments (see https://anonymous.4open.science/r/CCL_Figure-1662/RGBNT201_Correlation.jpg) confirm a significant negative correlation between training loss and the correspondence degree, establishing loss as an effective proxy for 'cognitive difficulty' in multi-modal tasks. Consequently, CDDO is not a blind sample-filtering scheme; rather, it leverages this correlation to drive a progressive cognitive evolution from easily aligned, strong-correspondence pairs to complex, weak-correspondence ones. This achieves a deep coupling between feature-level semantic refinement (CGSR) and optimization-level dynamic reweighting (CDDO).
>
>
> **Reply to Question (1)**: Thanks for your valuable question. Compared to the conventional Cross-Attention Mechanism, the Correspondence-Guided Semantic Refinement (CGSR) offers a critical advantage by transforming a "blind" feature aggregation into a "correspondence-aware" selective integration. Standard Cross-Attention treats all textual tokens as equally valid, which potentially aggregates irrelevant or noisy semantics from weak-correspondence pairs. Instead, CGSR incorporates a Semantic Consistency Mask that acts as a spatial-semantic filter, by explicitly modeling the reliability of image–text associations at the feature level, CGSR dynamically suppresses conflicting textual signals and prioritizes well-aligned attributes. This ensures that the fused representations are not corrupted by the inherent "noise" of MLLM-generated text, resulting in a more robust and discriminative multi-modal embedding than the unconstrained attention approach.
>
> **Reply to Question (2)**:
>  Thank you for the reviewer's question. Figure 5 visualizes the operational mechanism of the CDDO module within the proposed CCL framework.
>
> Figures 5(a) and 5(b) illustrate the relationship between the dynamic weights (y-axis) assigned by the model to different samples and the sample losses during training. The key pattern lies in the distribution of weights across samples:
>
> **Early Training (Figure 5(a)):** The weight distribution is highly concentrated. Only a small number of samples (corresponding to strong image-text pairs) receive high weights, while the vast majority of samples (corresponding to medium and weak correspondence pairs) are significantly suppressed to near zero. This reflects the model's initial "conservative strategy," prioritizing learning from the most reliable strong-correspondence samples.
>
> **Late Training (Figure 5(b)):** The weight distribution becomes more moderate. The range of samples receiving significant weights expands, with more previously suppressed medium and even some weak-correspondence samples beginning to gain effective weights and participate in optimization. This indicates that the model has developed stronger capabilities to progressively integrate more complex semantic associations.
>
> **Relationship with Correspondence Degrees:** The core driving factor behind the above phenomenon is the "image-text correspondence degree" of samples. The CDDO module uses sample loss as a proxy indicator for correspondence degree (smaller loss typically implies stronger correspondence). In the early stage, strong-correspondence (easy) samples have small losses, hence large weights; weak-correspondence (hard) samples have large losses, hence small weights. As the model learns, the losses of some medium-correspondence samples decrease, and their weights correspondingly increase. Therefore, the dynamic reallocation of weights is essentially a process where the model adaptively adjusts its attention to samples of different correspondence degrees based on learning progress.

---

> > ### Author Rebuttal · Reviewer_oL7b · 2026-04-03
> >
> > I thank the authors for the rebuttal, which has resolved my concerns. The added empirical evidence demonstrates a negative correlation between training loss and image-text correspondence. This provides a clear, data-driven justification for using loss as a proxy for the degree of image-text correspondence.
> >
> > The proposed CCL framework stands out for its technical rigor and logical consistency, with CGSR and CDDO jointly addressing the challenge of varying image-text correspondence at both the feature level and the optimization level. This design helps mitigate the impact of unreliable text on the joint embedding space. In my view, this overall strategy has a clear advantage in maintaining training stability, which is well supported by the strong experimental results.
> >
> > I maintain my positive score and strongly recommend acceptance. This paper is well-motivated and offers significant contributions to the multi-modal object ReID community.

---

### Official Review · Reviewer_du3N · 2026-03-06

**Soundness:** 3
**Presentation:** 4
**Significance:** 3
**Originality:** 3
**Overall Recommendation:** 5
**Confidence:** 4

**Summary:**

This paper proposes a new Correspondence Cognitive Learning (CCL) method for multi-modal object ReID. It models the correspondence degree between images and texts and facilitates a progressive learning process from easy to hard pairs. The authors propose a Correspondence-Guided Semantic Refinement (CGSR) module, which dynamically refines visual features with textual semantics according to the correspondence difficulty at the feature level. For sample pairs with different correspondence degrees, the authors further design a Cognitive-Driven Dynamic Optimization (CDDO) module that applies a self-paced weighting mechanism to adaptively adjust the optimization focus, thereby addressing the optimization-level correspondence challenge. By designing CGSR and CDDO, CCL learns more robust and more discriminative multimodal representations.

**Compliance With Llm Reviewing Policy:**

Affirmed.

**Ethics Expertise Needed:**

["Other Expertise"]

**Key Questions For Authors:**

See the weaknesses.

**Limitations:**

The proposed CCL leverages the correspondence degree between different modalities to enable a progressive learning process from easy to hard. This idea appears to be quite general and may be applicable to more tasks where paired samples exhibit varying degrees of correspondence. It is suggested that the paper further discuss the potential application scenarios of this method and its extensibility to other related tasks.

**Strengths And Weaknesses:**

Strengths：
1. The paper proposes a well-motivated method, CCL, which explicitly models the correspondence degree between images and texts and enables a progressive learning paradigm that moves from easy to hard sample pairs. Through this design, CCL achieves superior performance.

2. CGSR performs correspondence-guided fusion at the feature level, dynamically refining visual features with textual semantics and thereby learning more discriminative representations.

3. The CDDO module employs an efficient self-paced weighting mechanism to enable a dynamic learning process from easy to hard samples, thereby preventing weak-correspondence pairs from misleading the model during the early stages of training.

4. The paper systematically validates the effectiveness of the proposed CCL method through both theoretical analysis and experimental evaluation. Extensive and comprehensive experiments are conducted, demonstrating the superior performance of CCL from multiple perspectives.


Weaknesses:

1. The content in Figure 2 is somewhat redundant and does not clearly reflect the details of the proposed method. It is recommended to revise and simplify the figure.

2. The proposed CCL method is built upon the DEMO framework. Could this make the overall model overly complex?

3. For some image–text pairs with weak correspondence or even incorrect correspondence, would it be better to ignore the textual modality and rely solely on the image?

4. It is recommended to discuss the extensibility of CCL. For example, can CCL be applied to other multi-modal object ReID methods that use large models to generate auxiliary text, such as IDEA? In addition, can CCL be extended to address the noisy correspondence problem?

---

> ### Author Rebuttal · Authors · 2026-03-31
>
> **Reply to Weakness (1)**:
> We completely agree with your assessment. Figure 2 currently contains too much redundant textual information and lacks a clear, intuitive visual flow. In our revision, we will completely redesign Figure 2 with the following improvements:
> **Simplification**: We will abstract the standard Transformer blocks (CA/SA/FFN) into cleaner icons to reduce clutter.
> **Emphasizing Data Flow**: We will explicitly highlight the flow of the difficulty indicator ($d_{i}$) from the Loss Bank to both the CGSR fusion gate and the CDDO weighting mechanism using distinct, color-coded arrows.
> **Visualizing the Mechanism**: We will add a simplified conceptual subplot showing how the semantic mask $S^m$ filters out noisy visual tokens, making the core motivation immediately apparent without needing to read the dense text.
>
> **Reply to Weakness (2)**:
> Thanks for your valuable question. We respectfully clarify that while our method builds upon the DeMo framework, the added complexity is marginal and well-justified by the significant performance gains. a) Zero Parameters for CDDO: The CDDO module serves as a dynamic optimization strategy and does not introduce any additional learnable parameters. b) Minimal Overhead for CGSR: Although the CGSR module incorporates textual data, it introduces a minor parameter increase of less than 15 MB. c) Significant Performance Gains: As detailed in Table 4, the full CCL model only marginally increases the total parameters (from 98.79M to 103.79M) and FLOPs (from 35.10G to 46.16G) compared to the baseline, **yet delivers substantial performance improvements**.
>
> **Reply to Weakness (3) (4)**:
> We appreciate the reviewer’s positive feedback on the extensibility of CCL. Our framework is built on a decoupled, modular design that ensures broad generalizability. Specifically, the CDDO module is a model-agnostic, loss-level optimization strategy that can be directly applied to other text-augmented ReID frameworks (such as IDEA) without any architectural modifications. Meanwhile, the CGSR module functions as an independent feature-refinement plug-in to denoise textual semantics before multi-modal fusion. This **"plug-and-play"** design allows CCL to easily empower various existing methods that leverage MLLMs for auxiliary text generation.
>
> Regarding the Noisy Correspondence issue, we conducted a rigorous stress test on the RGBNT100 dataset. By randomly shuffling 20% and 50% of the textual annotations in the training set, the results demonstrate that CCL exhibits remarkable robustness due to its adaptive perception of correspondence degrees and the automatic down-weighting mechanism of CDDO. For instance, CCL maintains a high mAP of 86.8% even under extreme 50% noise. This strongly proves that CCL not only handles natural weak-correspondence pairs but also showcases its significant potential for solving the broader multi-modal noisy correspondence problem.
>
> | Dataset (rate) | mAP  | Rank-1 | Rank-5 | Rank-10 |
> |:----:|:----:|:------:|:------:|:-------:|
> |RGBNT100 (20%)  | 87.2 | 96.7   | 96.9   | 97.1    |
> |RGBNT100 (50%)  | 86.8 | 96.6   | 96.9   | 97.0    |
> |DeMo (baseline)  | 86.4 | 95.5   | 96.0   | 96.2    |

---

> > ### Author Rebuttal · Reviewer_du3N · 2026-04-03
> >
> > After reviewing all the comments and rebuttals, I think all my concerns have been well addressed. I will keep my positive score and recommend the acceptance of this paper.

---

### Official Review · Reviewer_VF4V · 2026-03-13

**Soundness:** 3
**Presentation:** 3
**Significance:** 3
**Originality:** 3
**Overall Recommendation:** 5
**Confidence:** 4

**Summary:**

This paper investigates the problem of using automatically generated text descriptions for assisted learning in multi-modal object re-identification. Existing methods typically use the generated text directly as a reliable supervision signal, but in reality, there are significant differences in the correspondence between images and text, and some descriptions may not perfectly match the semantics of the images, thus affecting model learning. To address this, the authors propose the Correspondence Cognitive Learning (CCL) framework, which improves the robustness of multimodal representation learning by explicitly modeling the correspondence between images and text.
Specifically, CCL comprises two key modules: Correspondence-Guided Semantic Refinement (CGSR) and Cognitive-Driven Dynamic Optimization (CDDO). CGSR selectively incorporates textual semantics at the feature level based on the difficulty of image-text correspondence, refining visual features through semantic consistency masks to suppress interference from unreliable text. CDDO, on the other hand, employs a self-paced dynamic weighting strategy at the optimization level, allowing the model to prioritize learning samples with high image-text correspondence and gradually incorporate more difficult samples for training. By jointly utilizing image-text correspondence information at both the feature modeling and optimization strategy levels, CCL can more robustly leverage text-assisted supervision.
Experimental results show that this method outperforms existing methods on multiple multimodal re-identification datasets (such as RGBNT201, RGBNT100, and MSVR310), validating the effectiveness of progressive learning using image-text correspondence cognition in improving multimodal re-identification performance.

**Compliance With Llm Reviewing Policy:**

Affirmed.

**Final Justification:**

Thank you for the detailed rebuttal. The authors have addressed my main concerns, and I consider the paper to be technically sound and well justified. Moreover, the proposed method improves the utilization of textual information by explicitly modeling image-text correspondence and enabling a progressive learning scheme from easy to hard pairs, making it both novel and insightful for multi-modal object ReID. I have accordingly increased my scores and support its acceptance.

One more suggestion: the additional experiment comparing inference with and without textual information, discussed in the response to the first question, is helpful for understanding the design choice. Including the corresponding quantitative results in a subsequent version of the paper would further improve clarity and completeness.

**Key Questions For Authors:**

1. Since the texture is not used during inference time. So, I wonder how it performs if we also use textual informations in during the inference time? Will it improve a lot?

2. It is confusing why the similarity score in Text-Guided Feature Refinement is based on the mixed visual feature, since it has already interacted with textual tokens, which may lead to uncorrect relation. Why not directly use original visual features?

3. I am curious about the quality of the generated textual descriptions. Have the authors evaluated their quality, and would using different large language models lead to significant differences in performance?

**Limitations:**

No. The paper does not provide a dedicated discussion on the limitations of the proposed method.

**Strengths And Weaknesses:**

Strengths:
- The idea of using textual descriptions in multi-modal ReID is interesting, and this work keeps the text only in the training process, which keeps a fair comparison with current methods.
- The motivation of the method and the design of its individual components are reasonable, and the ablation studies further demonstrate the effectiveness of the proposed approach.

Weakness:
- The training process is somewhat complex. As shown in Table 4, although the parameter improvement is not significant, the FLOPs improvement is substantial.
- The illustration in Figure 2 is not very clear. It does not effectively convey the functionality of the individual modules, and it is difficult to understand the authors’ intended workflow from the figure alone. I had to rely heavily on the textual description to grasp the purpose of the proposed components. Therefore, I strongly suggest that the authors redesign or improve this figure to better illustrate the framework.

---

> ### Author Rebuttal · Authors · 2026-03-31
>
> **Reply to Weakness 1**:
> We appreciate the reviewer pointing this out. The increase in FLOPs during training (from 35.10G to 46.16G) is primarily introduced by the Cross-Attention (CA) and Self-Attention (SA) operations within the CGSR module, as calculating attention matrices across sequence lengths inherently incurs higher computational costs. However, we want to emphasize a crucial advantage of our framework: this computational overhead is strictly limited to the training phase. Because the textual modality is only used as an auxiliary supervision signal during training (in accordance with standard visual ReID paradigms), the entire text encoder and the CGSR module are discarded during inference. Consequently, the inference FLOPs and parameter count of our CCL model are identical to the DeMo baseline (35.10G FLOPs and 98.79M Params).
>
> **Reply to Weakness 2**:
> We completely agree with your assessment. Figure 2 currently contains too much redundant textual information and lacks a clear, intuitive visual flow. In our revision, we will completely redesign Figure 2 with the following improvements:
> Simplification: We will abstract the standard Transformer blocks (CA/SA/FFN) into cleaner icons to reduce clutter.
> Emphasizing Data Flow: We will explicitly highlight the flow of the difficulty indicator ($d\_{i}$) from the Loss Bank to both the CGSR fusion gate and the CDDO weighting mechanism using distinct, color-coded arrows.
> Visualizing the Mechanism: We will add a simplified conceptual subplot showing how the semantic mask $S^m$ filters out noisy visual tokens, making the core motivation immediately apparent without needing to read the dense text.
>
> **Reply to Question 1**:
> This is a very insightful question regarding the potential of multi-modal inference. To address this, we conducted an additional experiment on the RGBNT201 dataset by incorporating textual descriptions during the inference phase. Specifically, we utilized the trained CGSR module to refine visual tokens with their corresponding textual features before calculating the final distance for retrieval.
> Experimental Analysis: Our results indicate that utilizing text during inference yields only a marginal performance improvement (0.1mAP) compared to our visual-only approach. We attribute this to the fact that during the training phase, the CGSR module effectively acts as a semantic scaffold, guiding the visual encoder to internalize high-level linguistic cues. Consequently, the visual backbone matures into a robust extractor capable of producing semantically enriched features independently. These findings further justify our primary focus on visual-only inference, as it achieves near-peak performance while avoiding the heavy computational overhead of generating accurate text descriptions for massive gallery sets in real-world deployments.
>
>
> **Reply to Question 2**:
> Rationality Analysis: The original visual tokens $\mathbf{F}\_v^m$ and textual embeddings $\mathbf{F}\_t^m$ reside in highly heterogeneous feature spaces. Directly computing the similarity between raw visual patches and text anchors often results in inaccurate and noisy correspondence scores, as the model struggles to bridge this large cross-modal gap in a single step.By first generating the mixed feature $\mathbf{F}\_{mix}^m$ via the initial Cross-Attention layer, we project the visual tokens into a "text-aware" contextual space. This step serves as a soft-alignment phase where visual features are enriched with global linguistic cues. Consequently, calculating the semantic mask $\mathbf{S}^m$ on $\mathbf{F}\_{mix}^m$ allows the subsequent similarity measurement (Eq. 3-4) to focus on fine-grained semantic verification rather than struggling with basic modality alignment. This hierarchical approach—initial soft-fusion followed by rigorous consistency verification—ensures a much more accurate and stable estimation of local semantic reliability, which is crucial for suppressing misleading signals in weak-correspondence samples.
>
> **Reply to Question 3**:
> We thank the reviewer for the insightful comment. Rather than directly evaluating or comparing the quality of texts generated by different large language models, our study focuses on reducing the model’s dependency on text quality. Therefore, in this work, we directly adopt the auxiliary texts provided by the IDEA method for training. The purpose of this design is to systematically investigate the impact of varying degrees of correspondence (i.e., strong vs. weak alignment) on the learning process under a fixed and realistically noisy textual condition.
>
> Experimental results show that, even under the same textual inputs, our method consistently outperforms IDEA, which validates the effectiveness and superiority of our approach in handling weak correspondence and noisy text.

---

> > ### Author Rebuttal · Reviewer_VF4V · 2026-04-02
> >
> > Thank you for the detailed rebuttal. The authors have addressed my main concerns, and I consider the paper to be technically sound and well justified. Moreover, the proposed method improves the utilization of textual information by explicitly modeling image-text correspondence and enabling a progressive learning scheme from easy to hard pairs, making it both novel and insightful for multi-modal object ReID. I have accordingly increased my scores and support its acceptance.
> >
> > One more suggestion: the additional experiment comparing inference with and without textual information, discussed in the response to the first question, is helpful for understanding the design choice. Including the corresponding quantitative results in a subsequent version of the paper would further improve clarity and completeness.

---

> > > ### Author Response · Authors · 2026-04-04
> > >
> > > We sincerely thank the reviewer for the positive feedback and support.
> > >
> > > The quantitative results of the experiment comparing inference with and without textual information can be seen in https://anonymous.4open.science/r/CCL_Figure-1662/RGBNT201_inference_design.png. These results further support our design choice and will be included in the revised paper.
> > >
> > > Thank you again for your valuable comments.

---

### Decision · Program_Chairs · 2026-04-30

**Decision:**

Accept (regular)

**Comment:**

This paper receives 3 accept recommendations and 1 weak accept recommendations, all reviewers' acknowledge the contribution of this paper.